# EEF1A1 deacetylation enables transcriptional activation of remyelination

Mert Duman ⬤ [1,2], Adrien Vaquié ⬤ [1], Gianluigi Nocera[1,2], Manfred Heller[3], Michael Stumpe[1], Devanarayanan Siva Sankar[1], Jörn Dengjel ⬤ [1], Dies Meijer ⬤ [4], Teppei Yamaguchi ⬤ [5], Patrick Matthias ⬤ [5], Thomas Zeis[6], Nicole Schaeren-Wiemers ⬤ [6], Antoinette Hayoz[1], Sophie Ruff[1] & Claire Jacob ⬤ [1,2✉]

Remyelination of the peripheral and central nervous systems (PNS and CNS, respectively) is a prerequisite for functional recovery after lesion. However, this process is not always optimal and becomes inefficient in the course of multiple sclerosis. Here we show that, when acetylated, eukaryotic elongation factor 1A1 (eEF1A1) negatively regulates PNS and CNS remyelination. Acetylated eEF1A1 (Ac-eEF1A1) translocates into the nucleus of myelinating cells where it binds to Sox10, a key transcription factor for PNS and CNS myelination and remyelination, to drag Sox10 out of the nucleus. We show that the lysine acetyltransferase Tip60 acetylates eEF1A1, whereas the histone deacetylase HDAC2 deacetylates eEF1A1. Promoting eEF1A1 deacetylation maintains the activation of Sox10 target genes and increases PNS and CNS remyelination efficiency. Taken together, these data identify a major mechanism of Sox10 regulation, which appears promising for future translational studies on PNS and CNS remyelination.

[1] Department of Biology, University of Fribourg, Chemin du Musée 10, 1700, Fribourg, Switzerland. [2] Department of Biology, Johannes Gutenberg University Mainz, Hanns-Dieter-Hüsch-Weg 15, 55128 Mainz, Germany. [3] Proteomics and Mass Spectrometry Core Facility, Department for BioMedical Research, University of Bern, Freiburgstrasse 15, 3010 Bern, Switzerland. [4] Center for Discovery Brain Sciences, Edinburgh Medical School, University of Edinburgh, 15 George Square, Edinburgh EH8 9XD, United Kingdom. [5] FMI for Biomedical Research, Novartis Research Foundation, Maulbeerstrasse 66, 4058 Basel, Switzerland. [6] Department of Biomedicine, University Hospital Basel, Hebelstrasse 20, 4031 Basel, Switzerland. ✉email: cjacob@uni-mainz.de

Many axons of the peripheral and central nervous systems (PNS and CNS, respectively) are myelinated. Two types of myelinating cells, Schwann cells (SCs) in the PNS and oligodendrocytes (OLs) in the CNS, build a thick myelin sheath rich in lipids around axons, which provides axonal insulation and saltatory nerve conduction[1]. Without myelin, the nervous system is poorly functional and sensitive to degeneration, thus myelin is critical for the function of the nervous system and the maintenance of its integrity. Different disorders in humans can lead to demyelination, hypomyelination, or dysmyelination. Demyelination occurs after traumatic lesions, which can regenerate in the PNS, but not in the CNS[2–4]. In the CNS, the most frequent demyelinating disease is an immune degenerative disease called multiple sclerosis that progresses through successive demyelinating lesions leading to permanent loss of function when remyelination fails[5–7]. In particular, the efficiency of CNS remyelination decreases dramatically with age, mainly due to a differentiation defect of OL precursor cells (OPCs) into mature myelinating OLs[5,6,8]. Other myelin disorders are inherited, such as subtypes of Charcot-Marie-Tooth disease[9–11], a large group of peripheral neuropathies, and the group of leukodystrophies in the CNS[12], or occur after an infection such as the Guillain-Barré syndrome[13]. Loss of myelin can also be secondary to diabetes, toxic agents, or aging. In summary, many pathological states can lead to myelin deficiency and thus to severe neurological disabilities. To optimize functional recovery and prevent irreversible loss of function, it is of utmost importance to identify treatments that enhance remyelination after lesion[14,15].

To enhance remyelination, it is crucial to understand the mechanisms controlling this process. PNS and CNS myelination mechanisms have been extensively studied and several transcription factors are known to be involved[16–18]. In addition, post-translational modifications including acetylation and methylation have been recently shown to hold critical functions in regulating transcription factor activity and gene expression during myelination and remyelination[19–21]. For instance, in OLs, the highly homologous histone deacetylases HDAC1 and HDAC2 (HDAC1/2) are required for differentiation and myelination[22]. Interestingly, HDAC1, HDAC3, and HDAC10 can promote nuclear localization of the transcription factor Olig1 by deacetylation and thereby OL maturation[23]. Methylation marks, such as trimethylation of histone H3 lysine 4 (H3K4me3), are also involved in the myelination process. Indeed, H3K4me3, an activating methylation mark, accumulates at enhancers of promyelinating factors such as *Myrf* and *Olig2* and of myelin genes to promote OL myelination[24,25]. In the PNS, HDAC4, together with HDAC3, deacetylate and silence the promoter of *cJun*, an inducer of SC demyelination, to promote myelination[26], however, the loss of HDAC3 leads to hypermyelination in adult mice and promotes remyelination after PNS lesion[27,28], suggesting a dual function for HDAC3 in the myelination and remyelination processes. The PRC2 complex, which adds repressive H3K27me3 marks, also possesses a dual function: it is necessary to prevent hypermyelination, whereas it also maintains myelin integrity by repressing injury-related genes[29]. Among the transcription factors that regulate myelination and remyelination, Sox10 holds major functions in both PNS and CNS[30–32]. Indeed, Sox10 activates the transcription of promyelinating and myelin genes such as *Oct6*, *Krox20,* and *Myelin protein zero* (*P0*) in SCs[33–35], and *Myrf* and *Myelin basic protein* (*Mbp*) in OLs[36,37]. The mechanisms that control the activity and expression of Sox10 remain however partially understood. We previously showed that HDAC1/2 act as co-factors of Sox10 in SCs and are thereby critical for SC development, myelination, and remyelination[38–41]. In the context of remyelination, HDAC2 allows the recruitment of a multifunctional protein complex containing Sox10 and the two H3K9 demethylases KDM3A and JMJD2C to *Oct6* and *Krox20* genes to de-repress these Sox10 target genes and allow their activation by Sox10[41]. In these previous studies, the direct deacetylation target of HDAC1/2 that enables the formation of this complex and thereby Sox10 activity remained however elusive. Here we show that Sox10 nuclear localization and expression are negatively regulated by eEF1A1, thereby controlling Sox10 activity. EEF1A (two highly homologous proteins eEF1A1 and eEF1A2 coded by two different genes) is described as a major translation elongation factor, however, several studies have identified additional functions for this factor[42]. Indeed, eEF1A has also been involved in the nuclear export of tRNAs, transcriptional regulation, proteolysis, apoptosis, oncogenesis, and viral propagation[42–44]. We demonstrate that the regulation of Sox10 by eEF1A1 is controlled by acetylation. Indeed, Ac-eEF1A1 translocates to the nucleus to interact with Sox10 and drag it out of the nucleus. We show here that HDAC2 deacetylates eEF1A1 in the nucleus, which induces its return to the cytoplasm and maintains Sox10 on its target genes. In addition, we report that theophylline, a known activator of HDAC2, decreases the levels of Ac-eEF1A1, increases the expression of Sox10 and of Sox10 target genes, and enhances the remyelination process in the PNS after a traumatic lesion and in the CNS after a demyelinating lesion in young adult and old mice.

## Results

**EEF1A1 is deacetylated by HDAC1/2 in SCs.** Sox10 is robustly downregulated at 1 day post sciatic nerve crush lesion (dpl) in adult mice (Fig. 1a), a well-characterized experimental model of PNS lesion where SCs rapidly demyelinate and convert into repair cells to promote axonal regrowth. At 12dpl, once axons have regrown, Sox10 is recruited to and activates its target gene *Krox20*[33,41], another major promyelinating factor[45,46], to induce the remyelination process, which is almost complete at 60dpl. We previously showed that HDAC1 and HDAC2, which can compensate for the loss of each other, are necessary for Sox10 functions in the PNS, and that HDAC2 is primarily involved in this process[38,39,41]. To identify direct targets of HDAC1/2, we collected at 1dpl sciatic nerves of adult mice treated with the HDAC1/2 inhibitor mocetinostat or its vehicle and analyzed the levels of acetylated proteins by mass spectrometry[41]. Interestingly, analysis by Protein Match Score Summation identified eEF1A1 as a putative HDAC1/2 deacetylation target (Supplementary Fig. 1a). To validate this, we ablated *Hdac1/2* specifically in SCs by crossing mice expressing the Cre recombinase under control of the *Dhh* promoter (*Dhh*-Cre[47]) with mice expressing floxed *Hdac1* and *Hdac2* (*Hdac1*fl/fl;*Hdac2*fl/fl[38,48]). We found strongly increased levels of Ac-eEF1A in SCs of postnatal day (P)4 *Dhh*-Cre;*Hdac1*fl/fl;*Hdac2*fl/fl knockout (dKO) nerves as compared to *Dhh*-Cre-negative control littermates (Fig. 1b, c), while total eEF1A1 levels were similar in control and dKO nerves (Fig. 1d), indicating that eEF1A deacetylation depends on HDAC1/2 in SCs. Consistently, HDAC1/2 inhibition using the HDAC1/2 inhibitor mocetinostat in primary SC cultures led to strongly increased levels of Ac-eEF1A1, but not of Ac-eEF1A2 (Fig. 1e, f and Supplementary Fig. 2). Mass spectrometry analyses revealed that all detected acetylated peptides of eEF1A1 were significantly more abundant in samples of mocetinostat-treated SCs as compared to vehicle-treated SCs (Supplementary Data 1). eEF1A1 has 20 putative acetylation sites[49]. Mass spectrometry analyses allowed us to detect 4 of these sites on K386, K219, K408, and K273, which are located in different domains of eEF1A1[49]. In addition, the antibody we used in Fig. 1b, c, e, f showing strongly increased levels of Ac-eEF1A in the absence of HDAC1/2 activity is directed against eEF1A acetylated at K41. These findings

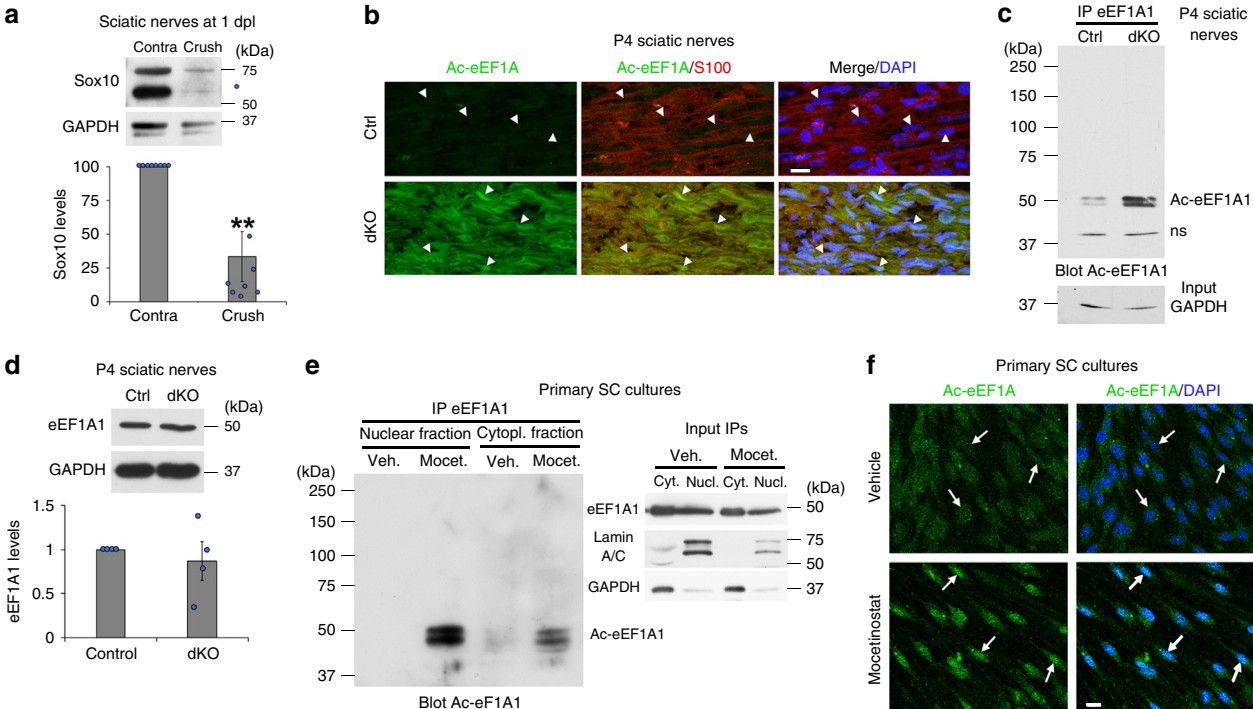

**Fig. 1 EEF1A1 is deacetylated by HDAC1/2 in SCs. a** Sox10 Western blot and quantification normalized to GAPDH showing low Sox10 levels at 1dpl in crushed as compared to contralateral sciatic nerves of adult mice. Paired two-tailed Student's $t$-tests, $p$ value = 0.008938, $n$ = 8 animals per group. **b**, **c** Co-immunofluorescence of Ac-eEF1A and S100 (SC marker) and DAPI (nuclei) labeling (**b**) and immunoprecipitation (IP) of eEF1A1 and Western blot of Ac-eEF1A (**c**) in sciatic nerves of P4 control (Ctrl) and DhhCre;Hdac1;Hdac2 knockout (dKO) mice showing increased levels of Ac-eEF1A in the absence of HDAC1/2 in SCs. n.s. = non-specific. **d** Western blot of total eEF1A1 and GAPDH (loading control) in sciatic nerves of P4 Ctrl and dKO mice and quantification of eEF1A1 levels normalized to GAPDH levels showing no significant difference between the two groups. Paired one-tailed Student's $t$-tests, $p$ value = 0.297774, $n$ = 4 animals per group. **e** IP eEF1A1 followed by Western blot of Ac-eEF1A in nuclear and cytoplasmic fractions of primary SCs cultured under de-differentiating conditions and treated with the HDAC1/2 inhibitor mocetinostat (Mocet.) or its vehicle (Veh.) for 3 days. Lamin A/C (nuclear marker), GAPDH (cytoplasmic marker) and eEF1A1 Western blots on lysates used for IP show the inputs. **f** Immunofluorescence of Ac-eEF1A and DAPI labeling in primary SCs cultured under de-differentiation conditions and treated with mocetinostat or its vehicle for 3 days. Arrowheads (**b**) and arrows (**f**) show SC nuclei. Z-series projections (**b**) or single optical sections (**f**) are shown. **b**, **f** Representative images of 3 different animals per group or of 3 independent experiments are shown. **b**, **d** Data are presented as mean values ± SEM. Scale bars: 10 µm. Source data are provided as a Source Data file.

suggest that eEF1A1 is globally acetylated in the absence of HDAC1/2 activity and that other lysine in addition to the ones detected by mass spectrometry and by the acetylK41-eEF1A antibody that we used here are most likely simultaneously acetylated in the absence of HDAC1/2 activity.

**Ac-eEF1A1 drags Sox10 out of the nucleus.** We found that eEF1A1 acetylation is increased in SCs upon de-differentiation in culture (Fig. 2a) and in sciatic nerves after lesion (Fig. 2b and Supplementary Fig. 1b). While eEF1A1 is mainly localized in the SC cytoplasm (Supplementary Fig. 3), we show that Ac-eEF1A1 is localized in both nuclear and cytoplasmic fractions (Fig. 1e, f). We found decreased nuclear levels of Sox10 in de-differentiated as compared to differentiated SCs and partial re-localization of Sox10 in the cytoplasmic compartment (Fig. 2c, d). In addition, Ac-eEF1A co-immunoprecipitated with Sox10 in both compartments, an interaction that was potentiated by HDAC1/2 inhibition in the nuclear compartment, which also led to a strong decrease of Sox10 levels in the nucleus (Fig. 2d). In line with its eEF1A1 deacetylating function, HDAC2 interacted with eEF1A1 in the nucleus of de-differentiated SCs in culture (Fig. 2e) and in sciatic nerves after lesion (Fig. 2b). We show that Sox10 is targeted to the proteasome in de-differentiated SCs (Fig. 3a and Supplementary Movie 1) and that inhibition of the proteasome results in increased Sox10 levels (Fig. 3b). Taking these data together, we hypothesized that eEF1A1 can translocate to the

nucleus to interact with Sox10, drag it out of the nucleus and target it to the proteasome for degradation. Indeed, overexpressed eEF1A1, which interacts with Sox10 in both cytoplasm and nucleus of de-differentiated SCs (Fig. 3c), leads to increased cytoplasmic re-localization of Sox10 (Fig. 3d) and increased co-localization of Sox10 with the proteasome in the cytoplasmic compartment (Supplementary Fig. 4) within 24 h, and to strongly decreased Sox10 levels within 3 days (Fig. 3e). In SCs cultured under de-differentiating conditions, approximately 25% of eEF1A1 is localized in the nucleus (Fig. 4a). To determine whether acetylation of eEF1A1 promotes its nuclear localization, we mutated individual lysine (K) into glutamine (Q) residues, which mimic acetylation. We chose K41, K179, and K273, which have been experimentally verified and several times described to be acetylated[49]. We found by subcellular fractionation that each mutation leads to increased nuclear localization of eEF1A1 (Fig. 4a). Among those, K273Q mutation was the most efficient. Consistently, the mutation of K273 into arginine (R), which prevents acetylation, was the most efficient to reduce nuclear localization of the protein (Fig. 4b). Interestingly, K273 is the only K residue specific to eEF1A1[49]. Indeed, in eEF1A2, K273 is replaced by R, thereby preventing acetylation at this position. Consistent with decreased nuclear localization of K273R mutant, overexpression of this mutant resulted in a decreased percentage of low Sox10-expressing SCs as compared to overexpression of wild-type eEF1A1 (Fig. 4c). Surprisingly, K179R mutation had a

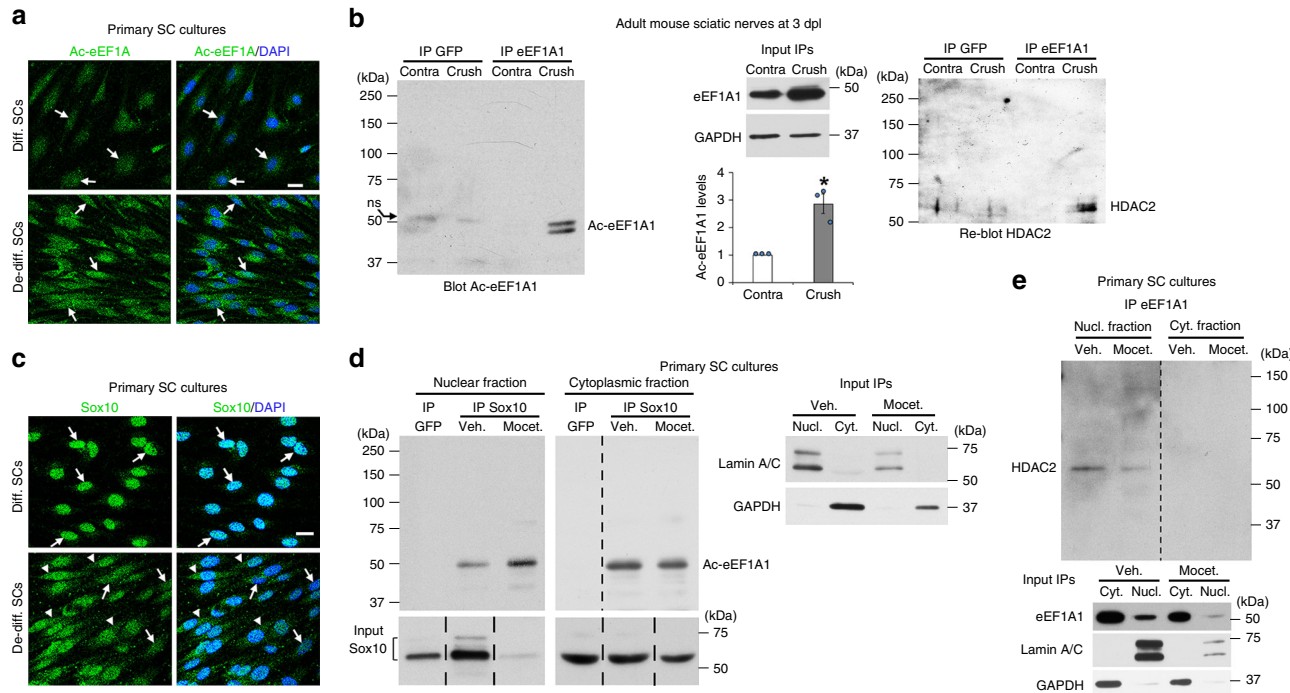

**Fig. 2 Sox10 re-localizes to the cytoplasm with Ac-eEF1A1 in de-differentiated SCs.** Immunofluorescence of Ac-eEF1A (**a**) and Sox10 (**c**), and DAPI (nuclei) labeling in differentiated and de-differentiated SCs. **b** Immunoprecipitation (IP) of eEF1A1 or GFP (control) carried out on the same pool of two adult mouse crushed or unlesioned contralateral sciatic nerves at 3dpl and Western blot of Ac-eEF1A followed by HDAC2. EEF1A1 and GAPDH Western blots on lysates show the input, n = 3 (6 animals). The graph shows the quantification of Ac-eEF1A levels (measured on a longer exposure to obtain a value for the contra IP eEF1A1) normalized to eEF1A1 input. Data are presented as mean values ± SEM. Paired two-tailed Student's t-tests, p value = 0.03376. IP Sox10 or GFP (**d**) or IP eEF1A1 (**e**) followed by Ac-eEF1A (**d**) or HDAC2 (**e**) Western blot in nuclear and cytoplasmic fractions of primary SCs cultured under de-differentiating conditions and treated with the HDAC1/2 inhibitor mocetinostat or its vehicle for 3 days. Lamin A/C (nuclear marker), GAPDH (cytoplasmic marker), Sox10 and eEF1A1 Western blots on lysates used for IP show the inputs. Dashed lines indicate that samples were run on the same gel but not on consecutive lanes. Arrows point to SC nuclei (**a**, **c**) and arrowheads (**c**) to SC cytoplasm. **a**–**e** Representative images of 3 independent experiments are shown. Scale bars: 10 μm. Source data are provided as a Source Data file.

similar effect as K273R on Sox10 levels (Fig. 4c). By analyzing Sox10 levels only in cells where eEF1A1 or the mutants were present in the nuclear compartment, we found that K179R prevents the decrease of Sox10 levels even when localized in the nucleus (Fig. 4d). In SCs cultured under proliferating conditions, overexpressed eEF1A1 is in a large majority found in the cytoplasmic compartment (Fig. 4e). We show in these conditions that the three K to Q mutations lead to increased nuclear localization of eEF1A1 (Fig. 4e) and to decreased Sox10 levels (Fig. 4f). We next tested which region of Sox10 protein is responsible for its decreased stability in the presence of Ac-eEF1A1. To this aim, we co-transfected HEK293 cells, which do not express Sox10, with either wild-type Sox10 of different Sox10 mutants together with K273Q eEF1A1 or with GFP as control. We show that expression of K273Q eEF1A1 leads to decreased levels of wild-type Sox10, Sox10-95 (mutant containing an insertion of 2 amino acids in the HMG domain, which impairs DNA binding[50]), Sox10-MIC (mutation resulting in a shorter protein containing the first 188 aminoterminal residues of Sox10, which include the HMG domain[50]), and Sox10-HMG (containing only the HMG domain[51]) (Supplementary Fig. 5a, b), indicating that the HMG domain is sufficient for Ac-eEF1A1 to decrease Sox10 stability. Similar to SCs, K273Q eEF1A1 co-immunoprecipitated with wild-type Sox10 in HEK293 cells (Supplementary Fig. 5c). In addition, K273Q eEF1A1 also co-immunoprecipitated with Sox10-95 (Supplementary Fig. 5c). These data show that Ac-eEF1A1 interacts with Sox10 in HEK293 cells such as in SCs and that the two additional amino acids in the HMG domain do not impair this interaction.

Taken together, these data indicate that acetylation of eEF1A1 at any of the three residues K41, K179 or K273 increases eEF1A1 nuclear localization and further decreases Sox10 levels and show that K179 and K273 are key acetylation sites for eEF1A1 nuclear localization and decrease of Sox10 levels.

**Tip60 associates with Stat3 to acetylate eEF1A1.** To understand the mechanism that leads to eEF1A1 acetylation after sciatic nerve crush lesion, we analyzed by mass spectrometry eEF1A1 binding partners in sciatic nerves at 2dpl compared to unlesioned nerves. EEF1A1 is a major translation elongation factor[42,44]; consistent with this, we detected several proteins involved in RNA transport and metabolism as eEF1A1 putative binding partners. Also consistent with a function of eEF1A1 interacting with the proteasome[52] and targeting proteins to the proteasome for degradation, we identified the proteasome regulatory proteins Proteasome activator subunits 1 and 2 as eEF1A1 putative binding partners (Supplementary Data 2). In addition, this analysis suggested that putative binding partners were recruited to eEF1A1 upon sciatic nerve crush lesion. Among those, Stat3 appeared as an interesting candidate to further investigate because it has been shown to interact with eEF1A1 in other cells[53] and with the histone acetyltransferase Tip60 in the cytoplasmic compartment[54], and is known to be expressed in SCs and activated after a sciatic nerve lesion[55–61]. We thus decided to further investigate the potential involvement of Stat3 and Tip60 in eEF1A1 acetylation. Consistent with previous findings[53,54], Tip60 and Stat3 co-immunoprecipitated with eEF1A1 (Fig. 5a) and Stat3 co-immunoprecipitated with Tip60 (Fig. 5b) in SCs. Tip60

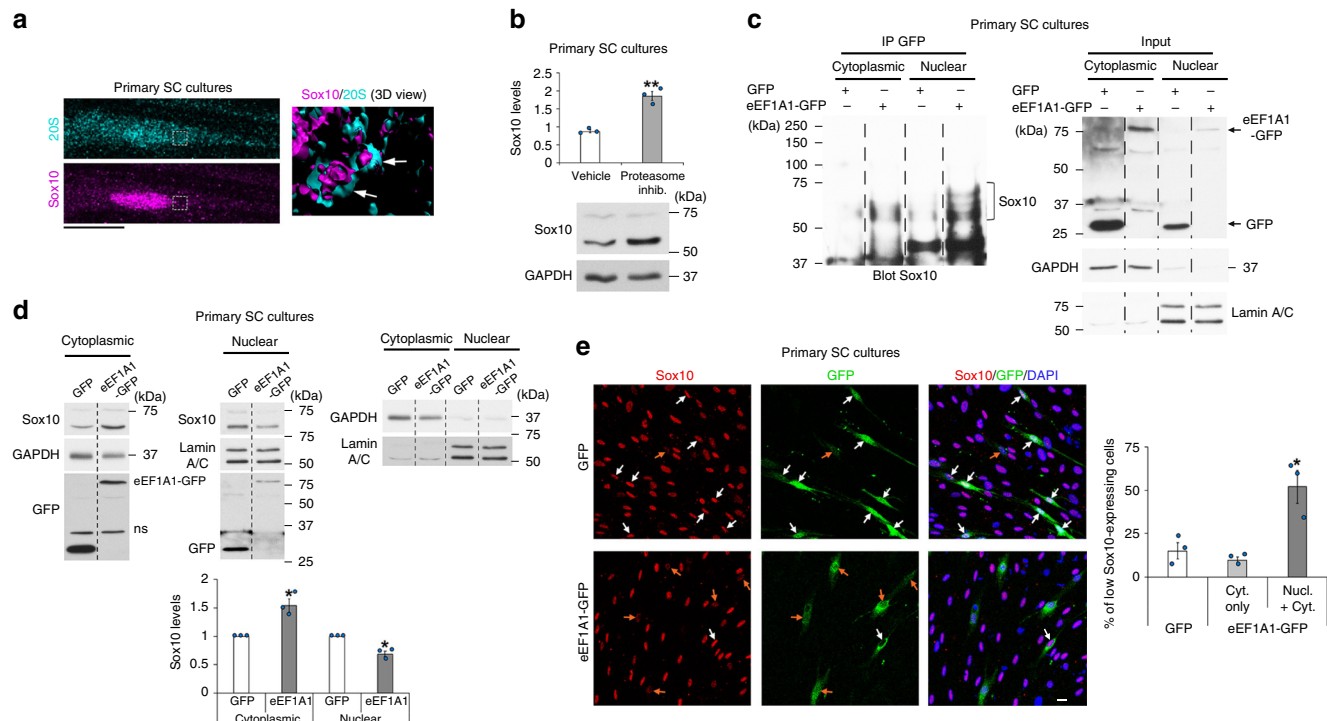

**Fig. 3 EEF1A1 re-localizes Sox10 to the cytoplasm and reduces its expression. a** Sox10 and 20 S co-immunofluorescence in SCs cultured under de-differentiation conditions. Representative images (z-series projections) are shown. The right image is a magnified 3D view of the region highlighted by a dashed box on the left images. Arrows point to proteasome structures containing Sox10. **b** Sox10 and GAPDH Western blot in lysates of SCs induced to de-differentiate for 1 day and treated with the proteasome inhibitor MG132 (proteasome inhib.) or vehicle (Veh.) for 12 h, and Sox10 quantification normalized to GAPDH ($n = 3$ independent experiments). **c** GFP immunoprecipitation (IP) and Sox10 Western blot in nuclear and cytoplasmic fractions of SCs transfected with eEF1A1-GFP or GFP and induced to de-differentiate for 1 day, and input GFP, LaminA/C (nuclear fraction) and GAPDH (cytoplasmic fraction) on lysates used for IPs. **d** Sox10 and GFP Western blots in nuclear and cytoplasmic fractions of SCs transfected with a construct expressing eEF1A1-GFP or control GFP and induced to de-differentiate for 1 day, and Sox10 quantification (normalized to GAPDH or Lamin A/C) in each fraction. n.s. = non-specific. **e** Sox10 immunofluorescence with GFP fluorescence and DAPI (nuclei) labeling in SCs expressing eEF1A1-GFP or control GFP and cultured under de-differentiating conditions for 3 days, and % of low Sox10-expressing cells among cells expressing GFP or eEF1A1-GFP in the cytoplasm only (Cyt. only) or in both nucleus and cytoplasm (Nucl. + Cyt.). Arrows point to transfected SCs (**e**). Orange arrows indicate transfected SCs with low Sox10 levels and/or with eEF1A1 localized in the nucleus (**e**). Dashed lines indicate that samples were run on the same gel but not on consecutive lanes. Data are presented as mean values ± SEM. Unpaired (**b**, **e**) or paired (**d**) two-tailed Student's t-tests. P values: 0.001725 (**b**), 0.045156 (**d**, cytoplasmic), 0.024327 (**d**, nuclear), 0.02473 (**e**). N = 3 independent experiments (**b**, **d**, **e**), 15–25 (GFP) and 48–58 (eEF1A1-GFP) transfected cells counted per n (**e**). **a–e** Representative images of 3 independent experiments are shown. Scale bars: 10 µm. Source data are provided as a Source Data file.

has a molecular weight of 60 kDa and the higher molecular weight isoforms of Tip60 from ~75 to ~100 kDa are sumoylated isoforms, which have been shown to have increased acetyl-transferase activity (Supplementary Fig. 6 and Refs. [62,63]). We show that eEF1A1 and Stat3 interact with sumoylated Tip60 in SCs (Fig. 5a, b). We found that Tip60 is mainly localized in the cytoplasmic fraction of SCs cultured under de-differentiating conditions (Fig. 5c). In addition, Tip60 is strongly upregulated after a sciatic nerve crush lesion during SC de-differentiation (Fig. 5d). We thus tested whether Tip60 is involved in eEF1A1 acetylation and Sox10 regulation in de-differentiated SCs. Indeed, inactivating Tip60 with the specific Tip60 inhibitor TH1834 robustly decreased eEF1A1 acetylation (Fig. 5e) and Tip60 binding to eEF1A1 (Fig. 5a), and increased Sox10 levels (Fig. 5f). Interestingly, downregulation of Stat3 by shRNA led to decreased levels of sumoylated Tip60 (Fig. 5g), decreased interaction of Tip60 with eEF1A1 (Fig. 5h) and decreased levels of Ac-eEF1A (Fig. 5i). Taken together, these results indicate that Tip60 associates with Stat3 to acetylate eEF1A1 in the cytoplasm of de-differentiating SCs early after a PNS lesion.

**Theophylline enhances PNS remyelination.** The mechanism of action of eEF1A1 on Sox10 and its regulation by HDAC2 are summarized in Fig. 6. We next aimed at using this mechanism to enhance remyelination in vivo. We show here that overexpression of wild-type eEF1A1 or of K273Q eEF1A1 in SCs cultured under differentiation conditions robustly decreases the percentage of Krox20-expressing SCs as compared to GFP-expressing SCs, and that expression of the K273Q mutant decreases further this percentage as compared to wild-type eEF1A1-overexpressing cells (Supplementary Fig. 7). *Krox20* is a Sox10 target gene and a major transcription factor for myelination in developing and adult SCs[45,46]. These data thus suggest that eEF1A1 interferes with the myelination program in SCs and shows that mimicking constitutive acetylation of K273 increases this function. We hypothesized that increasing the deacetylation rate of eEF1A1 by increasing HDAC2 expression and activity would prevent Sox10 cytoplasmic re-localization and degradation and thereby maintain the activation of Sox10 target genes (Fig. 6). Sox10 directly binds to and activates the promyelinating factor genes *Oct6* and *Krox20* in SCs and *Myrf* in OLs and myelin genes such as *P0* in SCs and *Mbp* in both SCs and OLs[30,32–34,36]. We previously showed that downregulation of HDAC2 in SCs by shRNA leads to decreased levels of Sox10, Krox20, and P0, while overexpression of HDAC2 increases Sox10, Krox20, and P0 levels[38]. Theophylline is a known pharmacological activator of HDAC2 at low dose,

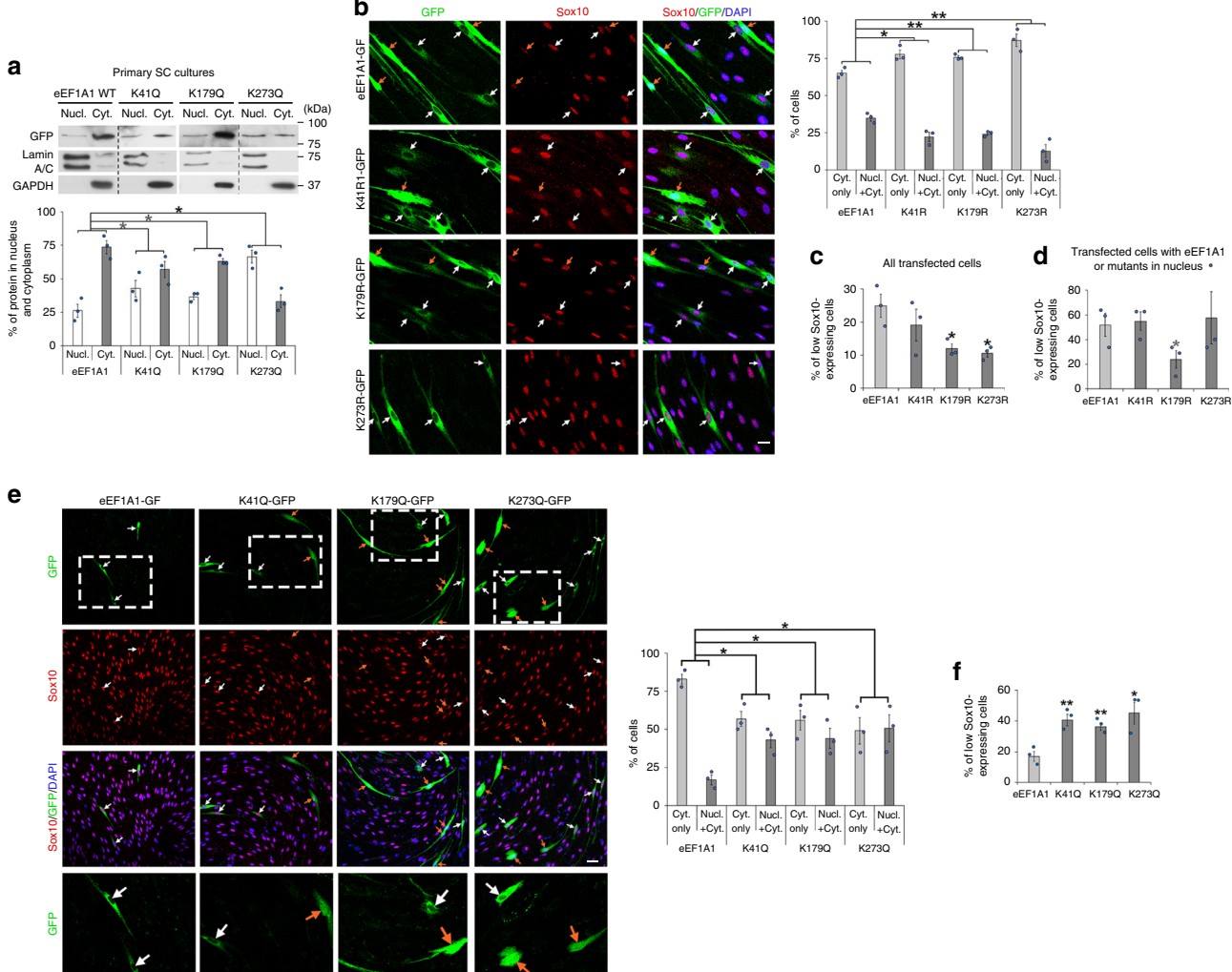

**Fig. 4 EEF1A1 acetylation increases eEF1A1 nuclear localization and decreases Sox10 levels. a** GFP Western blot in nuclear (Nucl.) and cytoplasmic (Cyt.) fractions of primary SCs cultured 1 day under de-differentiating conditions and transfected with eEF1A1-GFP, K41Q-GFP, K179Q-GFP or K273Q-GFP and % of protein localized in nucleus or cytoplasm normalized to Lamin A/C and GAPDH (n = 3 independent experiments per group). **b–d** Sox10 immunofluorescence with GFP fluorescence and DAPI (nuclei) labeling in SCs overexpressing eEF1A1-GFP, K41R-GFP, K179R-GFP or K273R-GFP and cultured under de-differentiating conditions for 3 days, and % of cells with eEF1A1 or mutants localized in cytoplasm only (Cyt. only) or in nucleus and cytoplasm (Nucl. + Cyt., **b**) or % of low Sox10-expressing cells (**c, d**). N = 3 independent experiments per group, 18–84 cells counted per group per n. **e, f** Sox10 immunofluorescence with GFP fluorescence and DAPI (nuclei) labeling in SCs overexpressing eEF1A1-GFP, K41Q-GFP, K179Q-GFP or K273Q-GFP and cultured under proliferating conditions for 2 days, and % of cells with eEF1A1 or mutants localized in cytoplasm only or in nucleus and cytoplasm of SCs (**e**) or % of low Sox10-expressing cells (**f**). The lower images are magnifications of the dashed white boxes on the upper images. N = 3 independent experiments per group, 17–71 cells counted per group per n. Arrows show transfected SCs. Orange arrows indicate SCs where eEF1A1 or the mutants are present in the nucleus of SCs. Scale bars: 10 μm (**b**), 20 μm (**e**). Data are presented as mean values ± SEM. Unpaired one-tailed (gray asterisks) or two-tailed (black asterisks) Student's t-tests, p values = 0.029415 (**a**, K41Q), 0.04884 (**a**, K179Q), 0.0329 (**a**, K273Q), 0.023698 (**b**, K41R), 0.008772 (**b**, K179R), 0.00946 (**b**, K273R), 0.189827 (**c**, K41R), 0.027448 (**c**, K179R), 0.018314 (**c**, K273R), 0.405588 (**d**, K41R), 0.03836 (**d**, K179R), 0.406686 (**d**, K273R), 0.011923 (**e**, K41Q), 0.019373 (**e**, K179Q), 0.021749 (**e**, K273Q), 0.009059 (**f**, K41Q), 0.009399 (**f**, K179Q), 0.02223 (**f**, K273Q). Source data are provided as a Source Data file.

increasing both HDAC2 activity and expression, while at higher concentrations theophylline acts as an antagonist of adenosine receptors and as an inhibitor of phosphodiesterases[64,65]. We show here that a low concentration of theophylline (1 μM) increases HDAC2 and Sox10 levels in primary SCs, whereas 1 μM CGS 15943 (antagonist of adenosine receptors) or 500 μM IBMX (inhibitor of phosphodiesterases) or 10 μM Rolipram (inhibitor of type IV phosphodiesterases) either have no effect or decrease HDAC2 and/or Sox10 levels (Supplementary Fig. 8a, b). At these concentrations, CGS 15943, IBMX, and Rolipram are fully active on their targets. These data show that theophylline does not increase Sox10 levels through inhibiting phosphodiesterases or

antagonizing adenosine receptors but is very likely to induce its effect on Sox10 through its third target HDAC2. Theophylline is already used as a drug in humans to treat asthma and chronic obstructive pulmonary disease[65]. Its activity, mode of administration and toxicity are thus very well described in mice and humans. To test whether theophylline enhances PNS remyelination, we carried out sciatic nerve crush lesions and treated mice with theophylline at 10dpl, just before SCs start remyelinating regenerated axons, for 2 or 4 consecutive days. We found that theophylline treatment indeed increases HDAC2 expression (Fig. 7a) and decreases Ac-eEF1A1 levels (Fig. 7b, c) as compared to vehicle-treated mice. Consistently, Sox10 levels were also

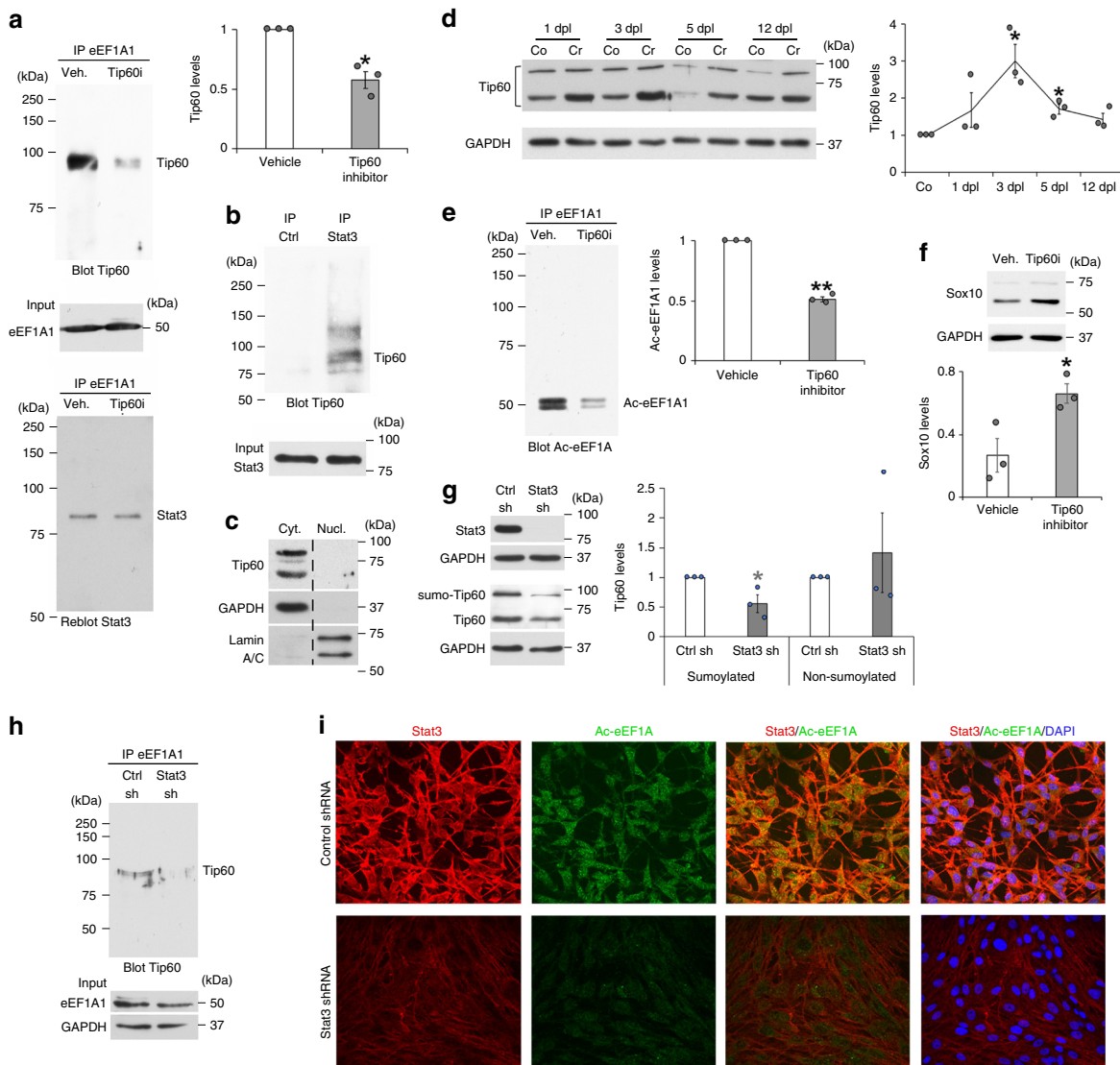

**Fig. 5 Tip60 acetylates eEF1A1 in a Stat3-dependent manner. a, e** eEF1A1 immunoprecipitation (IP) in lysates of SCs induced to de-differentiate for 1 day and treated with TH1864 (Tip60 inhibitor, Tip60i) or vehicle, and Tip60, Stat3 (**a**) or Ac-eEF1A (**e**) Western blot and quantification (normalized to eEF1A1) of Tip60 co-immunoprecipitated with eEF1A1 (**a**) or of Ac-eEF1A1 (**e**) in Tip60 inhibitor- compared to vehicle-treated SCs. **b** IP Stat3 or Flag (Ctrl) in lysates of SCs induced to de-differentiate for 1 day, Tip60 Western blot and Stat3 inputs. **c** Tip60, GAPDH (cytoplasmic marker) and Lamin A/C (nuclear marker) Western blots in cytoplasmic and nuclear fractions of SCs induced to de-differentiate for 1 day. Dashed lines: samples run on the same gel, but not on consecutive lanes. **d** Tip60 and GAPDH Western blots in lysates of crushed (Cr) and contralateral (Co) mouse sciatic nerves at 1-3-5-12dpl, and Tip60 quantification normalized to GAPDH in Cr compared to Co. **f** Sox10 and GAPDH Western blots in lysates of SCs induced to de-differentiate and treated with Tip60i or vehicle for 3 days and Sox10 quantification normalized to GAPDH. **g** Stat3 and Tip60 Western blots in SCs transduced with Stat3-specific shRNA (Stat3 sh) or non-targeting control shRNA (Ctrl sh) lentiviruses and quantification of sumoylated (higher molecular weight) and non-sumoylated (lower molecular weight) Tip60 isoforms normalized to GAPDH. **h** eEF1A1 IP and Tip60 Western blot in SCs transduced with Stat3 sh or Ctrl sh lentiviruses. **i** Stat3 and Ac-eEF1A co-immunofluorescence and DAPI labeling in primary SCs transduced with Stat3 sh or Ctrl sh lentiviruses and induced to de-differentiate for 2 days. Scale bar: 20 μm. Data presented as mean values ± SEM. Unpaired (**f**) or paired (**a, d, e, g**) one-tailed (gray asterisk) or two-tailed (black asterisks) Student's t-tests. P values = 0.029197 (**a**), 0.139771 (**d**, 1dpl), 0.047935 (**d**, 3dpl), 0.030298 (**d**, 5dpl), 0.061845 (**d**, 12dpl), 0.001796 (**e**), 0.033551 (**f**), 0.047979 (**g**, sumoylated), 0.300207 (**g**, non-sumoylated). N = 3 independent experiments or 3 animals per time-point. For each panel, representative images of 3 independent experiments are shown. Source data are provided as a Source Data file.

increased by theophylline treatment (Fig. 7a), which also led to increased levels of Krox20 and P0 (Fig. 7a). Sox10 is recruited to its target genes *Krox20* (at the myelinating SC element, MSE) and *P0* (at the intron 1) at the SC redifferentiation stage to induce remyelination (Fig. 7d). We show here that theophylline rapidly promotes the recruitment of Sox10 to its target genes (Fig. 7e), whereas the HDAC1/2 inhibitor mocetinostat impairs Sox10 recruitment (Fig. 7f), consistent with our previous findings showing impaired remyelination after lesion in sciatic nerves

lacking HDAC1/2 in SCs[41]. To test whether theophylline also increases Sox10 levels during developmental myelination, we treated mouse pups with theophylline from P1 to P3 and collected their sciatic nerves at P4. While theophylline had no effect on the levels of Sox10, P0 or HDAC2 in *Dhh*-Cre dKO mice lacking HDAC1/2 in SCs, we show that theophylline increases the levels of Sox10, P0, and HDAC2 in control littermate (*Dhh*-Cre negative) mice (Fig. 7g), indicating that theophylline can increase Sox10 and P0 levels also during development and suggesting that

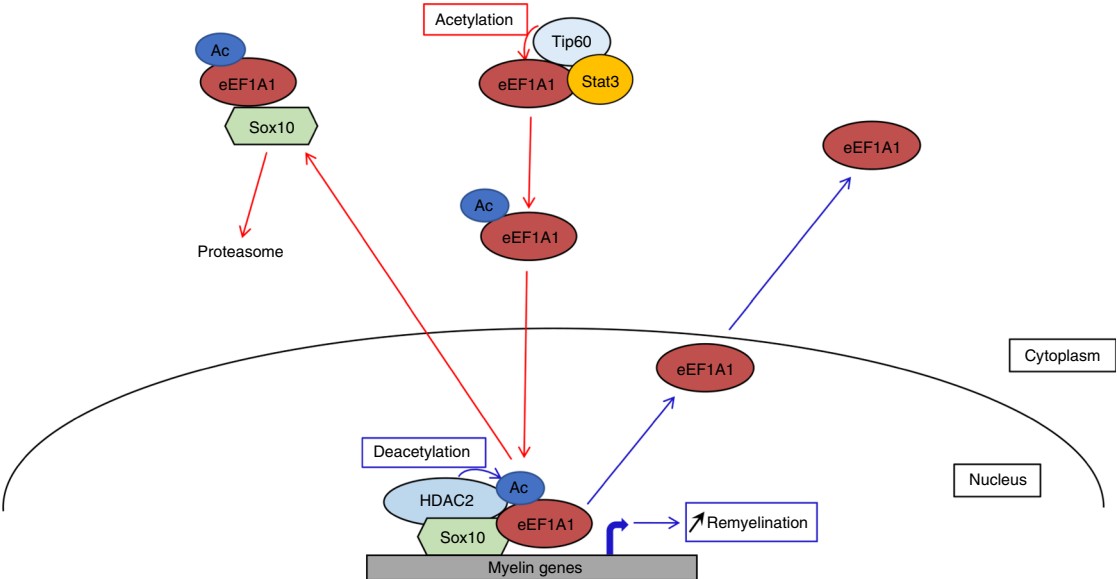

**Fig. 6 Proposed mechanism of Ac-eEF1A1-dependent regulation of Sox10.** Schematic representation of mechanisms controlling eEF1A1 acetylation-dependent Sox10 degradation and suggesting deacetylation-dependent remyelination.

this effect may require the presence of HDAC2 and/or HDAC1, although we cannot exclude a potential inability of the *Dhh*-Cre dKO mutants to express Sox10 and P0 with or without theophylline or that theophylline may act through additional or other pathways to increase Sox10 and P0 in control mice. In addition to increasing the expression of HDAC2, Sox10, Krox20, and P0, theophylline also increased myelin thickness (Fig. 8a) and led to faster motor and sensory functions recovery (Fig. 8b), while there was neither a difference in the density of SCs or of inflammatory cells (Supplementary Fig. 9a) nor a difference in the percentage of proliferating cells (Supplementary Fig. 9b). Consistently, theophylline treatment did not alter the density of SCs in culture (Supplementary Fig. 9c). Remyelinating sheaths are known to remain thinner as compared to myelin sheaths of unlesioned nerves, however, myelin in the injured nerve of theophylline-treated mice, but not of vehicle-treated mice, recovered a similar thickness as in uninjured nerves (Fig. 8c), indicating that theophylline allows full remyelination after lesion. Taken together, these data show that theophylline improves PNS remyelination efficiency after lesion.

**Ac-eEF1A1 regulates Sox10 levels in OLs.** Similar to SCs, eEF1A deacetylation in OLs depends on HDAC1/2 activity, as shown by a robust increase of Ac-eEF1A in OLs of mouse spinal cords 24 h after intrathecal injection of the HDAC1/2 inhibitor mocetinostat (Fig. 9a) and in primary OLs treated with mocetinostat (Fig. 9b). In addition, inactivating Tip60 with the specific Tip60 inhibitor TH1834 strongly reduced the increase of Ac-eEF1A due to mocetinostat treatment (Supplementary Fig. 10), suggesting that Tip60 acetylates eEF1A1 also in OLs. Such as in SCs, treatment with mocetinostat led to cytoplasmic relocalization of Sox10 and to strongly decreased Sox10 levels in primary OLs (Fig. 9c). Consistently, ablation of HDAC1/2 in mature OLs using the tamoxifen-inducible *PLP*-CreERT2 mouse line led to increased interaction of Ac-eEF1A with Sox10 (Fig. 9d) and to decreased Sox10 levels (Fig. 9e, f). To test whether eEF1A1 has a similar effect on Sox10 levels in OLs as in SCs, we overexpressed wild-type eEF1A1 or K273Q mutant in the oligodendroglial Oli-neu cell line[66] that can be easily transfected. We and others have shown that Oli-neu cells express high levels of MBP when cultured in differentiating conditions for 3 days and acquire a complex morphology after 10 days (Supplementary Fig. 11a and Ref. [66]). Consistent with our findings in SCs, K273Q mutant was localized in both nuclear and cytoplasmic compartments of virtually all transfected cells, while wild-type eEF1A1 was either localized exclusively in the cytoplasm or in both nucleus and cytoplasm of Oli-neu cells cultured under proliferating conditions (Supplementary Fig. 11b). We show here that overexpression of wild-type eEF1A1 or expression of K273Q mutant both decrease Sox10 levels as compared to control GFP transfection (Supplementary Fig. 11c, d) and that expression of K273Q mutant further decreases Sox10 levels as compared to overexpression of wild-type eEF1A1 (Supplementary Fig. 11c). Consistently, over-expression of wild-type eEF1A1 or expression of K273Q mutant both decreased MYRF levels as compared to control GFP transfection (Supplementary Fig. 11e, f) and expression of K273Q mutant further decreased MYRF levels as compared to over-expression of wild-type eEF1A1 (Supplementary Fig. 11e). Of note, K273Q mutant is re-localized to the cytoplasm of Oli-neu cells upon induction of differentiation (Supplementary Fig. 11b), possibly by a compensatory mechanism to recover Sox10 expression in the nucleus and thereby allow the induction of differentiation. To test whether K273Q mutant indeed impairs the induction of differentiation, we carried out luciferase gene reporter assays of the *Mbp* promoter 5 min after the induction of differentiation, where the activity of the *Mbp* promoter is already increased as compared to proliferating conditions (Supplementary Fig. 11g), in Oli-neu cells expressing K273Q mutant or GFP or overexpressing wild-type eEF1A1. While overexpression of wild-type eEF1A1 led to a trend of decreased activity of the *Mbp* promoter, expression of K273Q mutant decreased very significantly the activity of the *Mbp* promoter (Supplementary Fig. 1h), indicating that K273Q mutant impairs the activation of the Sox10 target gene *Mbp* and thus suggesting that K273Q mutant indeed impairs the induction of differentiation. At a later time-point in the differentiation process when MBP is already robustly expressed in Oli-neu cells (Supplementary Fig. 11a), we show that overexpression of wild-type eEF1A1 also strongly decreases the activation of the *Mbp* promoter (Supplementary Fig. 11i). Taken together, these data show that Ac-eEF1A1 is deacetylated by HDAC2 and negatively regulates the expression of Sox10 and of the Sox10 target genes *Myrf* and *Mbp* in OLs.

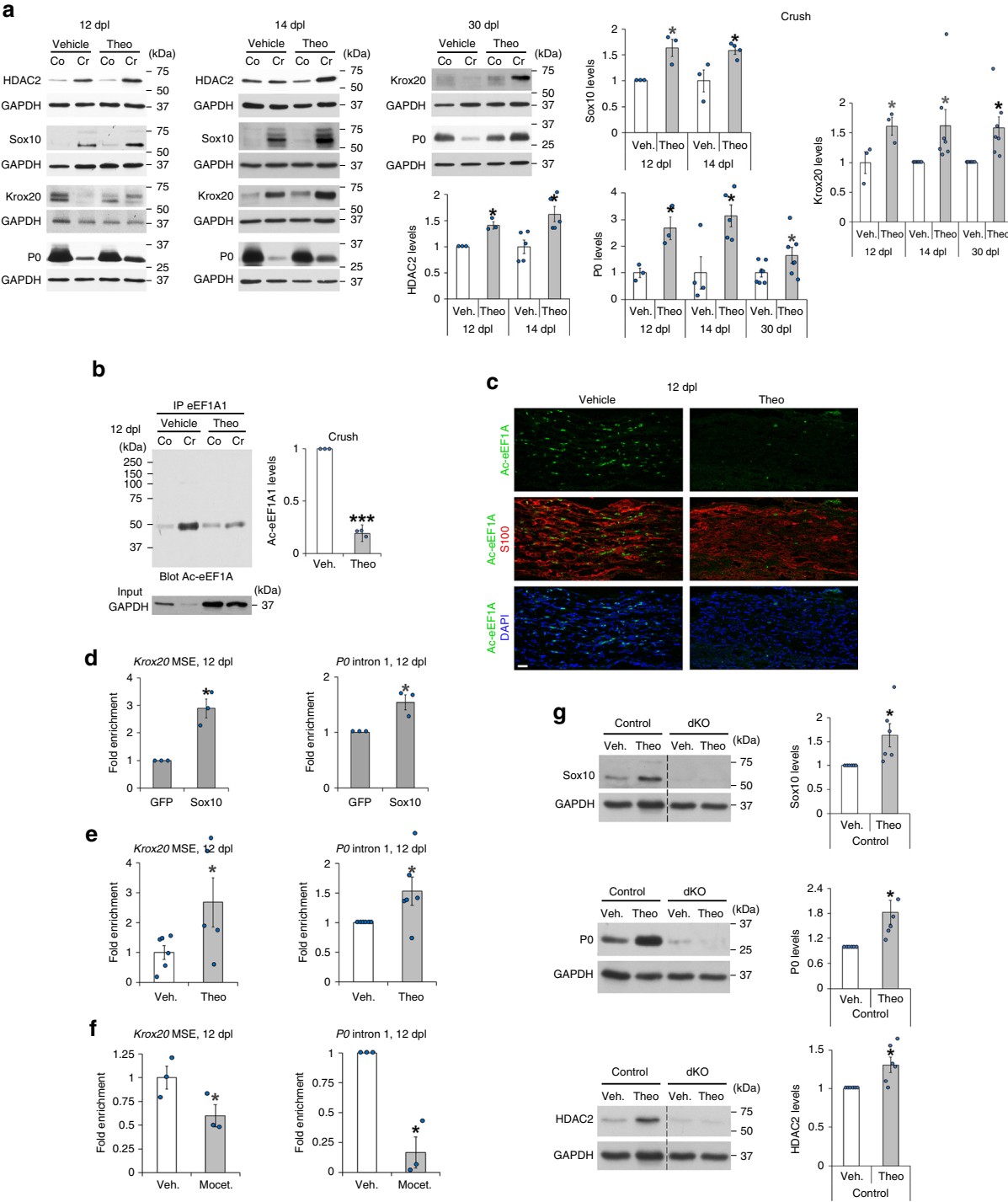

**Theophylline enhances CNS remyelination**. To test whether theophylline enhances CNS remyelination, we carried out focal demyelinating lesions in the spinal cord of adult mice using lysolecithin injection, a widely used and well-characterized experimental model of CNS demyelinating lesion where the remyelination process starts at 14 days post demyelinating lesion (dpdl). We show that a short treatment with theophylline from 10 to 14dpdl leads to a strong increase of HDAC2, Sox10 and MBP expression (Fig. 9g) and to improved remyelination efficiency at 14 and 30dpdl in young adults (Fig. 9h) and also in old mice (Fig. 9i), where remyelination efficiency is lower as compared to young adults[15]. Theophylline induced remyelination in the lesion site but did not affect myelination outside the lesion site, as

shown by unchanged g ratio (Fig. 9i, j). These data indicate that theophylline improves CNS remyelination efficiency after lesion.

## Discussion

Demyelination occurs in various cases in humans. In many cases, the remyelination process is not efficient enough and the loss of myelin leads to permanent loss of function. For this reason, it is crucial to better understand how remyelination is controlled in order to identify strategies that improve remyelination. In this study, we have focused our analyses on the remyelination process that occurs in the PNS after a traumatic lesion and in the CNS after a demyelinating lesion. In particular, we identified a mechanism that controls the activity and expression of Sox10, a

**Fig. 7 Theophylline increases Sox10, Krox20, and P0 expression and eEF1A1 deacetylation. a** HDAC2 (12dpl: $n = 3$, $p = 0.025883$, 14dpl: $n = 5$, $p = 0.011697$), Sox10 (12dpl: $n = 3$, $p = 0.031422$, 14dpl: $n = 3–4$, $p = 0.032671$), Krox20 (12dpl: $n = 3$, $p = 0.030486$, 14dpl and 30dpl: $n = 7$, $p = 0.033493$ and $0.022041$) and P0 (12dpl: $n = 3$, $p = 0.022239$, 14dpl: $n = 4$-5, $p = 0.019084$, 30dpl: $n = 6$, $p = 0.0456$) Western blots at 12, 14 and 30dpl in lysates of crushed (Cr) or contralateral (Co) sciatic nerves of mice treated at 10dpl with theophylline (Theo) or vehicle (Veh.) for 2 days (12dpl) or 4 days (14 and 30dpl), and quantification normalized to GAPDH and Co. **b, c** EEF1A1 IP and Ac-eEF1A Western blot (**b**) and Ac-eEF1A and S100 (SC marker) co-immunofluorescence and DAPI labeling (**c**) at 12dpl in Cr (**c**) or in Cr and Co (**b**) 20 h after one theophylline or vehicle injection, and (**b**) Ac-eEF1A1 quantification normalized to GAPDH input and Co in theophylline- compared to vehicle-treated groups. $N = 3$ animals per group (**b, c**), $p = 0.000597$ (**b**). Representative images are shown. **d–f** GFP (**d**) and/or Sox10 (**d–f**) chromatin immunoprecipitation at 12dpl on *Krox20* MSE or *P0* intron 1 in (**d**) Cr and fold enrichment normalized to Sox10 input compared to GFP ($n = 3$ animals, Sox10 and GFP IPs on the same nerve lysate, p*Krox20*-MSE = 0.032616, p*P0*-intron-1 = 0.027702) or in Cr (**e**) collected 20 h after theophylline (Theo) or vehicle (Veh.) treatment or (**f**) after mocetinostat (Mocet.) or Veh. treatment for 2 days. **e** $N = 5$-6 animals per group, p*Krox20*-MSE = 0.030466, p*P0*-intron-1 = 0.04003. **f** $N = 3$ animals per group, p*Krox20*-MSE = 0.037396, p*P0*-intron-1 = 0.023672. **g** Sox10, P0, HDAC2, and GAPDH Western blots in lysates of P4 *DhhCre;Hdac1*fl/fl;*Hdac2*fl/fl dKO and control littermate sciatic nerves treated at P1 with Theo or Veh. for 3 days, and quantification normalized to GAPDH in Theo compared to Veh. $N = 6$ animals per group, pSox10 = 0.04567, pP0 = 0.033358, pHDAC2 = 0.02891. Dashed lines: samples loaded on the same gel but not on consecutive lanes. Data are presented as mean values ± SEM. Paired (**a**: HDAC2 and Sox10-12dpl, Krox20-14dpl-30dpl; **b, d, e**, and **f**: *P0*-intron-1; **g**) or unpaired one-tailed (gray asterisks) or two-tailed (black asterisks) Student's *t*-tests. Scale bar: 20 μm (**c**). Source data are provided as a Source Data file.

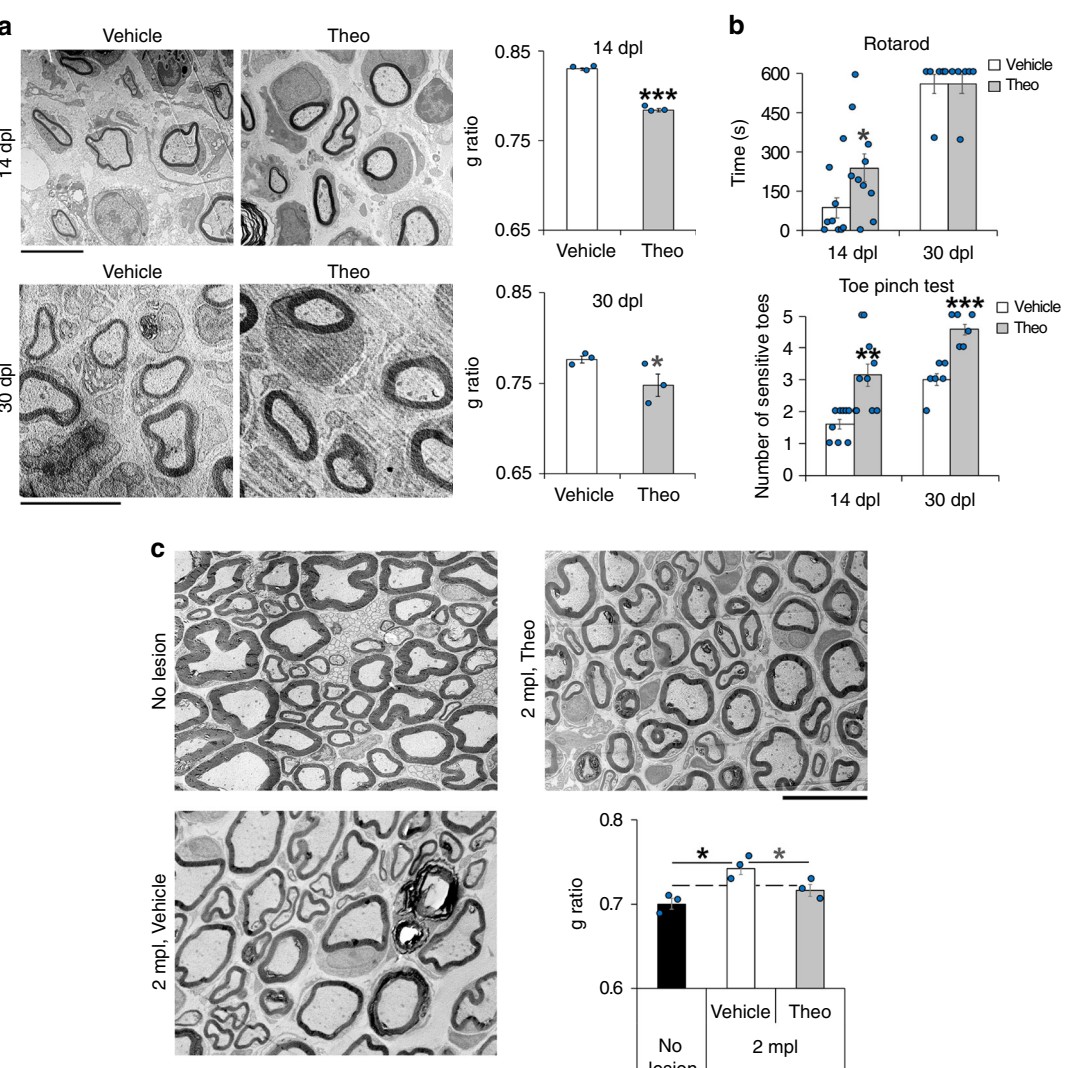

**Fig. 8 Theophylline accelerates and enhances PNS remyelination and functional recovery. a, c** Electron micrographs of Cr ultrathin sections at 14dpl, 30dpl (**a**) and 2mpl (**c**) from mice treated with theophylline or vehicle at 10dpl for 4 days, and myelin thickness quantification (g ratio) of remyelinated axons ($n = 3$ animals per group per time-point, 14dpl: $p = 0.000046$, 30dpl: $p = 0.048585$, 2mpl: pVehicle/No-lesion=0.013591, pTheo/No-lesion=0.076193, pTheo/Vehicle=0.034469). **b** Quantification of motor (Rotarod) and sensory (toe pinch) function recovery at 14 and 30dpl in theophylline- and vehicle-treated mice (4-day treatment at 10dpl). 14dpl: $n = 9$-10, pRotarod=0.02518, pToe-pinch=0.002327, 30dpl: $n = 6$ per group, pRotarod = 0.489629, pToe-pinch=0.000363. Data are presented as mean values ± SEM. Unpaired one-tailed (gray asterisks) or two-tailed (black asterisks) student's *t*-tests. Scale bars: 5 μm (**a, b**), 10 μm (**c**). Source data are provided as a Source Data file.

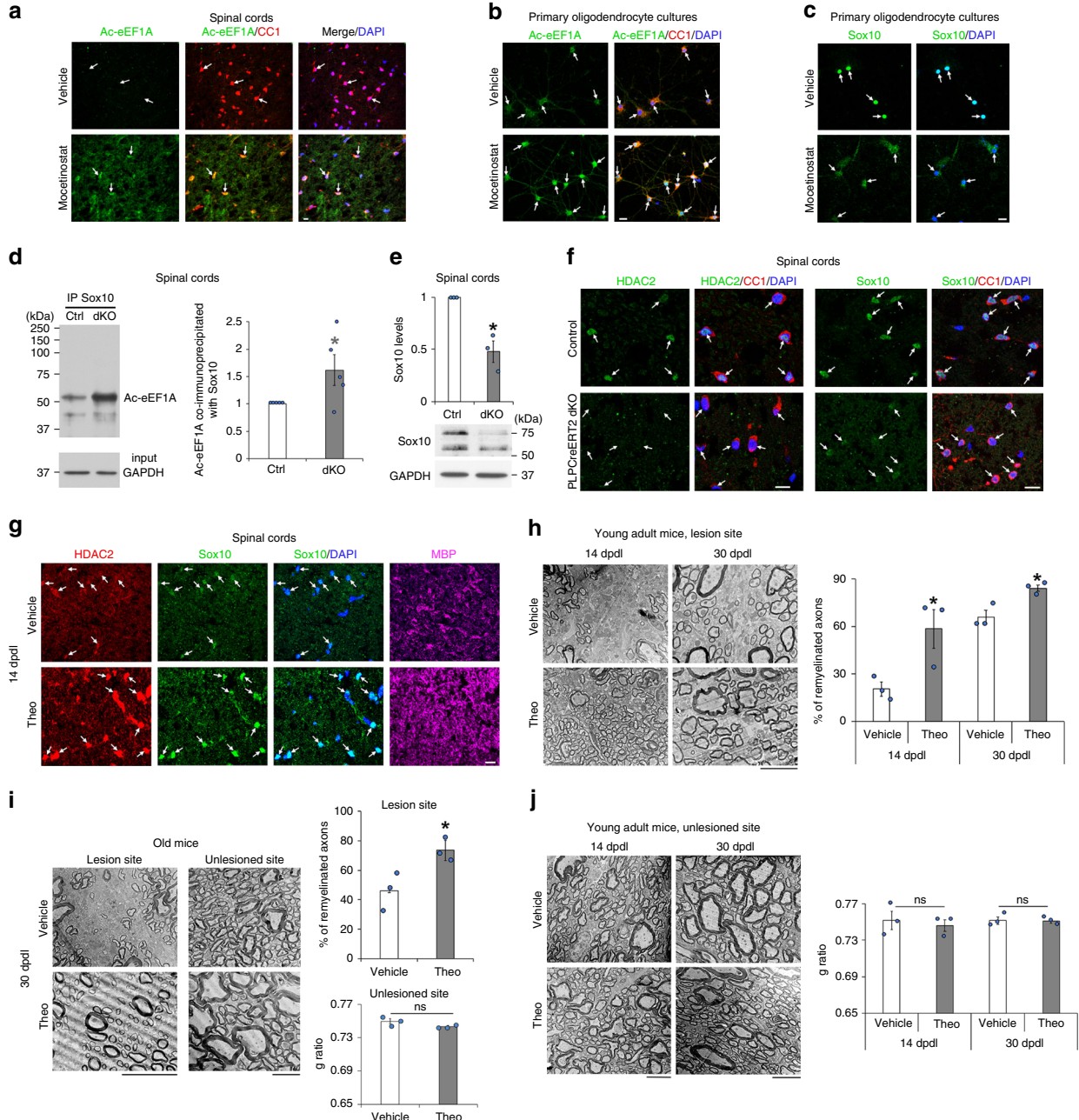

**Fig. 9 Theophylline promotes CNS remyelination in young and old adults. a** Co-immunofluorescence of Ac-eEF1A and CC1 (marker of mature OLs) and DAPI labeling in white matter of adult mouse spinal cord 24 h after treatment with mocetinostat or its vehicle. **b, c** Co-immunofluorescence of Ac-eEF1A (**b**) or Sox10 (**c**) with CC1 and DAPI labeling in differentiated primary OLs treated with mocetinostat or its vehicle for 24 h. **d–f** Co-immunoprecipitation (IP) of Ac-eEF1A with Sox10 (**d**) or Sox10 Western blot (**e**) and quantification normalized to GAPDH, or co-immunofluorescence of HDAC2 or Sox10 with CC1 and DAPI labeling (**f**) in spinal cords of *PLPCreERT2;Hdac1fl/fl;Hdac2fl/fl* (dKO) mice or control littermate (Ctrl) at 14 days post tamoxifen injections. **g** Co-immunofluorescence of HDAC2, Sox10 and MBP and DAPI labeling in the spinal cord lesion site of adult mice (3–4-month old) at 14dpdl after a 4-day treatment with theophylline (Theo) or its vehicle. Arrows indicate OLs. **h–j** Electron micrographs of spinal cord ultrathin sections in the lesion site (**h, i**) or unlesioned site (**i, j**) at 14 and/or 30dpdl in young adult (4-month old) (**h, j**) or old (19-month old) (**i**) mice, and % of remyelinated axons in the lesion site (delineated by axons with thin myelin, g ratio ≥ 0.835) or g ratio in unlesioned site. Data are presented as mean values ± SEM. Unpaired (**h, i, j**) or paired (**d, e**) one-tailed (gray asterisk) or two-tailed (black asterisks) Student's *t*-tests, p values: 0.047387 (**d**), 0.036845 (**e**), 0.043182 (**h**, 14dpl), 0.018162 (**h**, 30dpl), 0.127958 (**i**, unlesioned), 0.03517 (**i**, lesion), 0.328856 (**j**, 14dpl), 0.447458 (**j**, 30dpl). **d** n = 5 animals per group, (**e**) n = 3 animals per group, (**h–j**) n = 3 animals per group per time-point. Representative images of 3 animals per group (**a, f, g**) or of 3 independent experiments (**b, c**) are shown. Scale bars: white = 10 µm, black = 5 µM. Source data are provided as a Source Data file.

major transcription factor of PNS and CNS myelination and remyelination. We previously demonstrated that HDAC1/2 act as co-factors of Sox10[38-41], but the direct deacetylation mechanism underlying this function remained unclear. We show here that

HDAC1/2 deacetylate eEF1A1, which prevents interaction of eEF1A1 with Sox10 and re-localization of Sox10 to the cytoplasm, thereby allowing Sox10 to activate its target genes and enabling remyelination. We found that after a sciatic nerve crush lesion,

Tip60 is upregulated and acetylates eEF1A1 in the SC cytoplasm, in a Stat3-dependent manner. Consistently, Stat3 has been previously shown to interact with cytoplasmic Tip60[54] and with eEF1A1[53] in other cells. We show here that Stat3 promotes the interaction of Tip60 with eEF1A1 and eEF1A1 acetylation. eEF1A1 is a major translation elongation factor, which is mostly found in the cytoplasmic compartment. However, eEF1A1 has been shown to have non-canonical functions and to be localized in the nuclear compartment in some cases[42–44]. EEF1A1 can be submitted to various post-translational modifications, including phosphorylation, acetylation, methylation, ubiquitination[49]. Phosphorylation, which is the most studied post-translational modification of eEF1A1, has been shown to regulate eEF1A1 activity[42]. In contrast, the function of eEF1A1 acetylation is not or poorly described. We show here that acetylation of eEF1A1 induces its re-localization to the nuclear compartment of SCs. There are 20 putative acetylation sites in eEF1A1[49]. We studied the functions of the three acetylation sites K41, K179, and K273 because they have been several times verified experimentally. In addition, K273 is found only in eEF1A1 and not in eEF1A2[49]. Our results indicate that acetylation at any of these sites promotes nuclear localization of eEF1A1 and the decrease of Sox10 levels in SCs, and suggest that K179 and K273 are key acetylation sites for these functions. By using mocetinostat, a specific inhibitor of HDAC1/2, or by ablating HDAC1/2 specifically in SCs or in OLs, we show that HDAC1/2 are necessary to deacetylate eEF1A1 in SCs and OLs, and to maintain high Sox10 expression in the nuclear compartment of these cells.

In the aim of identifying a treatment that promotes this mechanism and thereby potentially improves remyelination efficiency, we tested the effect of theophylline treatment in vivo. Although theophylline inhibits phosphodiesterases and antagonizes adenosine receptors when used at high concentration, it is also a known potent activator of HDAC2, increasing HDAC2 activity and expression when used at low concentration[64,65]. We show here that while theophylline increases the levels of HDAC2 and Sox10 in SCs, compounds that specifically inhibit phosphodiesterases or antagonize adenosine receptors either have no effect on HDAC2 and Sox10 levels or decrease the levels of HDAC2 and Sox10, indicating that theophylline does not increase Sox10 levels through its effect on phosphodiesterases or adenosine receptors but is highly likely to act on Sox10 through HDAC2. Interestingly, a short treatment with theophylline in mice increased HDAC2 expression, decreased Ac-eEF1A1 levels, increased the expression of Sox10 and its target genes and improved remyelination efficiency in both PNS and CNS. Of major importance, theophylline also increased the efficiency of CNS remyelination in old mice, thereby suggesting a potential beneficial effect on the remyelination process in aged individuals, where remyelination is strongly impaired.

We previously showed that in differentiated primary SCs where Sox10 levels are already high and which express the major promyelinating factor Krox20 and the myelin protein P0, the overexpression of Sox10 results in a 5-fold induction of the Krox20 MSE[38], a critical enhancer for Krox20 expression. In addition, overexpression of HDAC2 alone leads to a 3-fold induction of the Krox20 MSE and co-overexpression of Sox10 and HDAC2 has a synergistic effect on the activation of the Krox20 MSE, with a 25-fold increase[38]. These data indicate that increased Sox10 levels in differentiated SCs can further increase the activation of its target gene Krox20 and that this is potentiated by simultaneous increase of HDAC2 expression. In the case of remyelination, Sox10 is upregulated at the remyelination stage to induce the remyelination program by the upregulation of Krox20 and of myelin proteins. We show here that theophylline treatment at 10dpl before remyelination starts allows to reach faster a high upregulation of Sox10, faster recruitment of Sox10 to its target genes Krox20 and P0, faster upregulation of Krox20 and P0, faster remyelination and faster functional recovery. We also showed that theophylline treatment does not change the density of SCs or of inflammatory cells nor the percentage of proliferating cells in the sciatic nerve after lesion. Upregulating Sox10 to a high level faster is thus very likely to induce faster remyelination and faster functional recovery. In addition, raising Sox10 to a higher expression level is also likely to help sustain the remyelination process for a longer time, as shown by the high expression of Krox20 and P0 at 30dpl and more efficient remyelination in theophylline-treated mice at 2 months post lesion. Thus, our previous work[38] and current findings suggest that Sox10 levels and activity including the timing of Sox10 upregulation and the levels of nuclear Sox10 are rate-limiting for the speed and efficiency of remyelination, and that increasing Sox10 levels and activity in conjunction with HDAC2 appears as a very promising strategy to accelerate and improve remyelination after lesion.

In summary, this study identifies Ac-eEF1A1 as a critical negative regulator of the remyelination process and shows that theophylline, by activating HDAC2, promotes eEF1A1 deacetylation, increases Sox10 levels and activity and remyelination speed and efficiency after lesion of the PNS and CNS, thus appearing as a very promising compound to test in future translational studies to accelerate and promote remyelination after traumatic lesions or in the context of demyelinating disorders.

## Methods

**Statistical analyses.** For each data set presented, experiments were performed at least three times independently or with at least three animals and p values were calculated in Microsoft Excel (Mac version 16.34) using two-tailed (black asterisks) or one-tailed (gray asterisks) Student's t-tests. P values: *<0.05, **<0.01, ***<0.001, data are presented as mean values ± SEM. Individual data points are represented in the graphs by scatter points. Sample size was determined by the minimal number of animals or individual experiment required to obtain statistically significant results and increased in some cases to improve confidence in the results obtained. No animal or data point was excluded from the analysis.

**Animals.** We ablated HDAC1/2 specifically in SCs by crossing mice expressing the Cre recombinase under control of the Dhh promoter[47] with mice expressing floxed Hdac1 and Hdac2[48]. To induce ablation of HDAC1/2 in OLs of adult mice, mice expressing floxed Hdac1 and Hdac2 were crossed with mice expressing tamoxifen-inducible Cre recombinase under control of the OL (and SC)-specific Plp promoter[67]. To ablate HDAC1/2, mice received daily injections of 2 mg tamoxifen (Sigma) for five consecutive days. Genotypes were determined by PCR on genomic DNA.

For all surgical procedures, we used isoflurane (3% for induction, 1.5-2% for narcosis during the operation) for anesthesia. For analgesia, 0.1 mg/kg/body weight buprenorphine (Temgesic; Essex Chemie) was administered by i.p. injection 1 h before surgery, a second time in the evening on the day of surgery and afterwards every 12 h during 3 days. The mice were placed on a heat pad during the entire procedure until waking up from anaesthesia. To prevent dehydration of the eyes, a carbomer liquid eye gel (e.g. Viscotears, Novartis) was used preoperatively. Mice were shaved either at the height of the hip for sciatic nerve crush lesion or on their back for lysolecithin lesion of the spinal cord and the field of operation was cleaned and disinfected. Sciatic nerve crush lesions were carried out on 3 to 4-month old adult mice (males and females) as follows (by a procedure that we have previously described[41]): An incision was made at the height of the hip and the sciatic nerve was exposed on one side. The nerve was crushed (5 ×10 sec with crush forceps: Ref. FST 00632-11). The wound was closed using Histoacryl Tissue Glue (BBraun). After the operation, mice were wrapped in paper towels until recovery from anaesthesia. Lysolecithin injections were carried out at T8 level as follows (by a procedure that was previously described[68]): to prevent dehydration, a single i.p. injection of 500 µl electrolyte solution with glucose (e.g. Aequifusine, B. Braun Medical) was administered preoperatively under anesthesia. A 1 cm long skin incision was made through the lower thoracic spine of the animal, followed by the separation of the paravertebral muscles from their insertion points at the processi spinosi over a length of ~3 mm to allow access to the spinal cord. The ligamentum flavum, which connects the dorsal lamella of two adjacent vertebrae, was then incised and the underlying dura mater was pierced with a needle at the injection site (right of the central vein). Two microliters of lysolecithin (1% in saline) were injected focally into the dorsal funiculus and the ventral horn of the spinal cord with a glass capillary. We used a stereotactic three-way micromanipulator to hold the syringe and insert it into the spinal cord at a defined angle (15 degrees from

vertical position) and at a defined depth: the needle of the Hamilton syringe was slowly inserted into the spinal cord on the right of the central vein at T8 vertebrae level until resistance was observed. The needle was then withdrawn 0.1 mm, 1 µl of lysolecithin was injected within 30 s, then the experimenter waited 30 s, the needle was then moved up 0.5 mm, 1 µl of lysolecithin was injected again within 30 s, the experimenter waited 1 min, then the needle was withdrawn.

Stratified random allocation in blocks with gender, age, and weight as strata was used. To minimize heterogeneity, we used littermate mice as control vehicle mice. Experimenters were blinded to the experimental group (genotype, treatment) and received only the animal number given at birth by the animal caretaker. Treatment allocation and collection/analysis of data were done by different persons. Treatments: 10 mg/kg mocetinostat (HDAC1/2 inhibitor) or 10 mg/kg theophylline or their respective vehicle were injected in the pelvic cavity after wound closure and treated again once a day for the indicated time in the figure legends. In mouse pups, 10 mg/kg theophylline or its vehicle were injected subcutaneously once a day for 3 days at P1, P2, and P3 and collection of tissues was carried out at P4. For analysis in the spinal cord of adult mice (young adults: 3–4-month old; aged mice: 18-month old; males and females), 10 mg/kg mocetinostat or its vehicle were injected intrathecally once and animals were sacrificed 24 h later, or 10 mg/kg theophylline or its vehicle were injected intraperitoneally once a day for 4 days at 10, 11, 12, and 13dpdl. All mice used in this study were from mixed strains backcrossed at least 10 times to C57BL/6 J. Mice were housed in a standard mouse facility with controlled ventilation (inward airflow, exhaust to the outside), temperature ($22 \pm 2°C$) and humidity ($65 \pm 5\%$) in individually ventilated type II long cages (L 365 mm × B 207 mm × H 140 mm) containing sawdust bedding, a cardboard cylinder and 2 paper tissues on the floor, food pellet and water ad libitum. Light cycle: 12:12 h. This study complies with all relevant ethical regulations concerning animal use, which was approved by the Veterinary office of the Canton of Fribourg, Switzerland and the Veterinary office (Landesuntersuchungsamt) of Rheinland-Pfalz, Germany.

**Functional recovery experiments.** Mice were placed four times on the Rotarod apparatus at a fixed speed of 15 revolutions per minute to test balance and motor coordination, at 14 and 30dpl. The duration of each trial was limited to 600 s, and trials were separated by a 30 min recovery period. Latency to fall from the rotating beam was recorded and the average of the three trials was used for quantification. Recovery of sensory function was tested at 14 and 30dpl by toe pinch test: each toe of the rear foot on the right side (lesioned side) was pinched with equal pressure applied by the same experimenter using flat tip forceps. Immediate withdrawal was recorded as functional sensitivity of the pinched toe. In case no toe exhibited sensitivity, the same test was applied to toes of the contralateral side (uninjured side), which always resulted in immediate withdrawal. All tests were carried out with the same experimental animals (10 theophylline-treated mice and 9 vehicle-treated mice).

**Constructs.** Expression constructs: pCMV6-AC-mouse eEF1A1-turbo GFP (C-ter) and GFP backbone constructs were acquired from Origene. Control shRNA (PLKO) and Stat3 shRNA (SHCLNG-NM_011486, TRCN0000071456) were acquired from Sigma-Aldrich. Sox10 wild type and mutant constructs, 95, MIC and HMG and the *Mbp* promoter-luciferase construct were kindly provided by Dr. Michael Wegner (University of Erlangen, Germany). K to Q or to R eEF1A1 mutants were generated by site-directed mutagenesis using pCMV6-AC-mouse eEF1A1-turbo GFP as template. Primers were as follows: K41Q forward, 5′-GGAATCGACAAGCGAACCATCGACAAGTTTGAGAAAGAGGCTGCTGA G - 3′, K41Q reverse, 5′- CTCAGCAGCCTCTTTCTCAAACTTGTCGATGGTTC GCTTGTCGATTCC - 3′, K41R forward, 5′- GGAATCGACAAGCGAACCATC GAAGAGTTTGAGAAAGAGGCTGCTGAG - 3′, K41R reverse, 5′- CTCAGCA GCCTCTTTCTCAAACTCTTCGATGGTTCGCTTGTCGATTCC - 3′, K179Q forward, 5′- CGTTAAGGAAGTCAGCACCTACATTCAGAAAATTGGCTACAA CCCTGACACAG - 3′, K179Q reverse, 5′- CTGTGTCAGGGTTGTAGCCAATTT TCTGAATGTAGGTGCTGACTTCCTTAACG - 3′, K179R forward, 5′- CGTTA AGGAAGTCAGCACCTACATTAGGAAAATTGGCTACAACCCTGACACAG - 3′, K179R reverse, 5′- CTGTGTCAGGGTTGTAGCCAATTTTCCTAATGTAGG TGCTGACTTCCTTAACG - 3′, K273Q forward, 5′- GCCGAGTGGAGACTG GTGTTCTCCAACCTGGCATGGTGGTTACCTTTGC - 3′, K273Q reverse, 5′- GCAAAGGTAACCACCATGCCAGGTTGGAGAACACCAGTCTCCACTCGG C - 3′, K273R forward, 5′- GCCGAGTGGAGACTGGTGTTCTCAGACCTGGC ATGGTGGTTACCTTTGC - 3′, K273R reverse, 5′- GCAAAGGTAACCACCA TGCCAGGTCTGAGAACACCAGTCTCCACTCGGC - 3′. All constructs were verified by sequencing.

**Cell culture.** Purified primary rat SC cultures were obtained as follows (method previously described[69]): primary SCs were derived from P2 Wistar rat sciatic nerves and dissociated in 0.3 mg/ml collagenase type I (Sigma-Aldrich) and 2.5 mg/ml trypsin (Sigma-Aldrich) in DMEM (Invitrogen) at 37 °C and 5% CO₂/95% air for 1 h. After the addition of DMEM containing 10% FCS (Gibco), cells were centrifuged at 500 × g for 10 min, resuspended in DMEM containing 10% FCS, 1:500 penicillin/streptomycin (Invitrogen), and 10 µM cytosine arabinoside (Sigma-Aldrich), and plated on plastic dishes coated with poly-D-lysine (Sigma-Aldrich). After 24 h at 37 °C and 5% CO₂/95% air, cells were washed and incubated at 37 °C

and 5% CO₂/95% air until they reached confluency in SC proliferating medium: DMEM containing 10% FCS (Gibco), 1:500 penicillin/streptomycin (Invitrogen), 4 µg/ml crude GGF (bovine pituitary extract, Bioconcept), and 2 µM forskolin (Sigma). SCs were then purified by sequential immunopanning in plastic dishes coated with a Thy1.1 antibody[70]. Identity and purity were checked for each primary preparation by immunofluorescence of SC-specific markers (p75, Sox10, Oct6, Krox20, P0, MAG). SCs were grown in proliferation medium: at 37°C and 5% CO₂/95% air. SC de-differentiation culture protocol mimicking SC demyelination and conversion into repair cells that occur after a PNS lesion was as follows (previously described[41]): SCs were first growth-arrested in defined medium (DM[58]) for 8 to 15 h, then 1 mM dbcAMP (Sigma) was added to induce differentiation. Cells were incubated in this medium for another 3 days. The medium was then changed to DM only without dbcAMP, and RSCs were incubated in this medium for 3 days (differentiation mimicking adult SC stage). To induce de-differentiation, cells were then changed to proliferation medium and incubated in this medium for 1 to 3 days (exact duration stated in the figure legends). For some experiments, 10 µM MG132 (proteasome inhibitor), 5 µM TH1834 (Tip60 inhibitor), 0.6 µM mocetinostat (HDAC1/2 inhibitor), 1 µM theophylline (HDAC2 activator at this concentration), 1 µM CGS 15943 (adenosine receptor antagonist), 500 µM IBMX (phosphodiesterase inhibitor), 10 µM rolipram (type 4 phosphodiesterase inhibitor) or their respective vehicle were added on cells.

Primary OLs were isolated from P1 to P2 rat cortices, which were first diced, incubated with DNase I and Trypsin for 15 min at 37°C and 5% CO₂/95% air, and then washed and centrifuged at 100 × g for 5 min. The pellet was then re-suspended and homogenized in DMEM20S[71], incubated 10 min on ice, passed through a 70-µm cell strainer and cultured on poly-D-lysine-coated flasks for 10 days in DMEM20S at 37 °C and 5% CO₂/95% air. The flasks were then tightly closed and shaken first for 1 h at 37 °C to remove microglial cells, and then overnight. OPCs were collected in the supernatant (this procedure has been previously described[71]). Identity and purity were checked for each primary preparation by immunofluorescence of Sox10 and CC1.

Oli-neu cells[66] (kind gift from Dr Jacqueline Trotter, University of Mainz, Germany) were cultured either in proliferating conditions in DMEM containing 1% horse serum (Gibco) and the following media components: 40 nM Biotin, 100 µg/ml Apo-transferrin, 16 µg/ml Putrescine, 6 ng/ml Progesterone, 4.8 ng/ml Sodium selenite, 5 µg/ml Insulin (all components from Sigma), 1:500 Penicillin-Streptomycin (Invitrogen), or in differentiating conditions in DMEM containing 100 µg/ml Apo-transferrin, 16 µg/ml Putrescine, 6 ng/ml Progesterone, 4.8 ng/ml Sodium selenite, 5 µg/ml Insulin, 40 ng/ml Triiodothyronine (all components from Sigma), 1:500 Penicillin-Streptomycin (Invitrogen) on poly-D-lysine (Sigma)-coated dishes.

HEK293 cells were obtained from ATCC and were cultured in DMEM containing 10% FCS (Gibco) and 1:500 penicillin/streptomycin (Invitrogen) on poly-D-lysine (Sigma)-coated dishes.

Mycoplasma contamination was not tested, because of the low incidence of mycoplasma contamination in primary cells, and because mycoplasma contamination results in inefficiency of transfection, which we did not observe in our primary cells or in Oli-neu cells or in HEK293 cells.

**Ultrathin sections and electron microscopy.** Mice were killed with 150 mg/kg pentobarbital i.p. (Esconarkon; Streuli Pharma AG) and sciatic nerves or spinal cords were fixed in situ (sciatic nerves) or by perfusion (spinal cords) with 4% paraformaldehyde (PFA) and 3% glutaraldehyde in 0.1 M phosphate buffer, pH 7.4. Fixed tissues were post-fixed in 2% osmium tetroxide, dehydrated through a graded acetone series[40], and embedded in Spurr's resin (Electron Microscopy Sciences). Ultrathin sections (70-nm thick) were cut. G-ratio (diameter of axon: diameter of axon+myelin) analyses were done at 12 mm distal to the lesion site for sciatic nerves and in the lesion site for spinal cords (g ratio average of 60–250 axons calculated per animal). To delineate the lesion site in the spinal cord and identify remyelinated axons, the g-ratio was calculated. Axons with a g-ratio above or equal to 0.835 were classified as remyelinated axons. This g-ratio threshold value was defined after calculating the g-ratio of axons in non-lesioned mouse spinal cords, where axons with the thinnest myelin sheath had a g-ratio under or equal to 0.83. No contrasting reagent was applied. Images were acquired using a Philips CM 100 BIOTWIN equipped with a Morada side-mounted digital camera (Olympus).

**Immunofluorescence on mouse tissues and cell cultures.** Mouse sciatic nerves were fixed in situ with 4% PFA for 10 min, dissected, embedded in O.C.T. Compound (VWR chemicals), and frozen at −80°C. Sciatic nerve cryosections (5-µm thick) were first incubated with acetone for 10 min at −20°C, washed in PBS/0.1% Tween 20, blocked for 30 min at room temperature (RT) in blocking buffer (0.3% Triton X-100/ 10% Goat serum/ PBS), and incubated with primary antibodies overnight at 4°C in blocking buffer. Sections were then washed 3 times in blocking buffer and secondary antibodies were incubated for 1 h at RT in the dark. Sections were washed again, incubated with DAPI for 5 min at RT, washed and mounted in Citifluor (Agar Scientific).

Cells were washed with PBS, fixed for 10 min with 4% PFA at RT, washed again in PBS, blocked, incubated with antibodies, DAPI, and mounted as described above.

For immunofluorescence on spinal cords, mice were deeply anesthetized with a lethal dose of pentobarbital and perfused with 4% PFA after blood removal with heparin. Spinal cords were collected 2 mm above and below the lesion site, post-fixed in 4% PFA for 2 h at RT, incubated in 20% sucrose overnight at 4°C, embedded in O.C.T. compound, and frozen at -80°C. We used 5-μm cryosections. Sections were first submitted to antigen retrieval in citrate buffer (10 mM citrate buffer, 0.05% Tween20, pH6) for 2 h at 65°C, washed, blocked, and incubated with antibodies, DAPI, and mounted as described above.

Primary antibodies: eEF1A1 (rabbit, 1:200, abcam, cat. # ab157455, lot # GR231741-13), eEF1A-pan (Acetyl-Lys41) (rabbit, 1:200, AssaybioTech, cat. # D12106, lot # 410212106), HDAC2 (rabbit, 1:100, Santa-Cruz Biotechnology, cat. # sc-7899, lot # E0912), Sox10 (rabbit, 1:200, DCS Innovative Diagnostik-Systeme, cat. # SI058C01, lot # S294), Sox10-Nterm (rabbit, 1:200, kindly provided by Dr. Michael Wegner), CC1 (mouse, 1:200, Calbiochem, cat. # OP80, lot # 2869730), Olig2 (goat, 1:200, R&D systems, cat. # AF2418, lot # UPA0512081), 20 S proteasome subunit (mouse, 1:200, abcam, cat. #ab22674, lot # GR283474-6), MBP (rat, 1:200, BIO-RAD, cat. #MCA409S, lot # 161031 A), S100 (mouse, 1:100, GeneTex, cat. # GTX11179, lot # 821503399), F4/80 (rat, 1:200, Lucernachem, cat. # GTX26640, lot # 821403776), STAT3 (mouse, 1:200, abcam, cat. #ab119352, lot # GR306889-4). All secondary antibodies were from Jackson ImmunoResearch. Photos were acquired using a Leica TCS SP-II confocal microscope or a Visitron VisiScope spinning disk confocal system CSU-W1. Fiji (version 1.0) and Adobe Photoshop (CC 20.0.8 Release) were used to process images. Single optical sections or z-series projections (stated in the figure legends) are shown.

**Chromatin immunoprecipitation**. We used a previously described protocol[29] with modifications: freshly dissected mouse sciatic nerves were minced in 1% formaldehyde for 10 min for crosslinking, which was then quenched for 10 min with glycine to a final concentration of 0.125 M. Samples were then sequentially lysed as follows: samples were first resuspended in ice cold lysis buffer 1 (50 mM HEPES, 140 mM NaCl, 1 mM EDTA, 10% glycerol, 0.5% NP-40, 0.25% TX-100, protease inhibitors), rotated at 4°C for 10 min and centrifuged at 4°C at 1350 x g for 5 min. After discarding the supernatant, the same procedure was repeated in lysis buffer 2 (10 mM Tris, 200 mM NaCl, 1 mM EDTA, 0.5 mM EGTA, protease inhibitors). The pellet was then resuspended in lysis buffer 3 (10 mM Tris, 100 mM NaCl, 1 mM EDTA, 0.5 mM EGTA, 0.1% Na-Deoxycholate, 0.5% N-Laurylsarcosine, 0.03% SDS, protease inhibitors), incubated on ice for 10 min, homogenized and diluted by 2 in lysis buffer 3. Chromatin was fragmented by sonication with a Bioruptor (Diagenode) set at medium power, interval: 30 s on / 45 s off, time: 4 cycles of 10 min, rest of 2 min between cycles. Sonicated chromatin containing 75 μg of proteins and 3 μg of antibodies were used per IP. Sonicated chromatin was incubated overnight a 4°C on a rotating wheel, and Magna ChIP™ Protein A + Protein G Magnetic Beads (Millipore) were added and incubated for 2 h at 4°C. Magnetic beads were washed, immunoprecipitated samples were eluted in elution buffer (50 mM Tris, 10 mM EDTA, 1% SDS), crosslinking was reversed at 65°C for 8–16 h and after RNase and proteinase K treatment, the DNA was purified using a PCR clean-up kit (Macherey-Nagel). Quantitative real-time PCR analyses were performed with an ABI 7000 Sequence Detection System (Applied Biosystems) using FastStart SYBR Green Master (Sigma), according to the manufacturer's recommendations. A dissociation step was added to verify the specificity of the products formed. Primers sequences were as follows: Krox20 MSE, forward, 5'-TTTCGTCTTTGGGGCTCATTC-3', reverse, 5'-AGCCCTTCACAAAGCTGAA A-3', P0 intron 1, forward, 5'-GTGCTGGACTCGGCATCTTT-3', reverse, 5'-CCAGGTTACCATGACTGGGG-3'. For Sox10 IP, we used 2 μg/IP of anti-Sox10 antibody (rabbit, DCS Innovative Diagnostik-Systeme, cat. # SI058C01) and as negative control IP we used 2 μg/IP of anti-GFP antibody (rabbit, Abcam, cat # ab290, lot # GR196475-1).

**Western blot and immunoprecipitation**. For all in vivo Western blots and immunoprecipitations, we have collected the injured sciatic nerve from the lesion site to around 12 mm distal to the lesion site (region where the nerve splits into the three branches of tibial, sural and common peroneal nerves). We have collected the same region of the contralateral nerve as internal control for each animal. After perineurium removal, sciatic nerves were frozen in liquid nitrogen, pulverized with a chilled mortar and pestle, lysed in radioimmunoprecipitation assay (RIPA) buffer (10 mM Tris/HCl, pH 7.4, 150 mM NaCl, 50 mM NaF, 1 mM NaVO4, 1 mM EDTA, 0.5% wt/vol sodium deoxycholate, and 0.5% Nonidet P-40) for 30 min on ice, and centrifuged to pellet debris. Supernatants were collected.

Cells were washed once in PBS, lysed in RIPA buffer for 15 min on ice, and centrifuged to pellet debris. Sciatic nerves and cell lysates were submitted to SDS-PAGE and analyzed by Western blotting.

Primary antibodies used: eEF1A1 (rabbit, 1:5000, Abcam, cat. # ab157455), eEF1A-pan (Acetyl-Lys41) (rabbit, 1:500, Labforce/AssaybioTech, cat. # D12106), HDAC2 (mouse, 1:1000, Sigma, cat. # H2663, lot # 096M4799V), Sox10 (rabbit, 1:250, DCS Innovative Diagnostik-Systeme, cat. # SI058C01), Sox10 (mouse, 1:1000, Abcam, cat. # ab216020, lot # GR3272630-2), GAPDH (glyceraldehyde-3-phosphate-dehydrogenase, mouse, 1:5000, Genetex, cat. # GTX28245, lot # 821705388), P0 (chicken, 1:1000, Aves Labs, cat. # PZO, lot # PZO0308), Krox20 (rabbit, 1:500, provided by Dr. Dies Meijer, University of Edinburgh, Ref. [72]), Lamin A/C (mouse, 1:5000, Sigma-Aldrich, cat. # SAB4200236, lot # 055M4822V),

KAT5/Tip60 (rabbit, 1:1000, abcam, cat. #ab151432, lot # GR113712-14), STAT3 (mouse, 1:5000, abcam, cat. #ab119352, lot # GR306889-4), GFP (mouse, 1:2000, Origene, cat. # TA150041, lot # W002).

All secondary antibodies were from Jackson ImmunoResearch: light chain-specific goat anti-mouse-HRP (horse radish peroxidase) and goat anti-rabbit-HRP, and heavy chain-specific goat anti-chicken-HRP.

For non-denaturing IPs, tissues were prepared and lysed as described above in the following RIPA buffer: 10 mM Tris/HCl, pH7.4, 150 mM NaCl, 50 mM NaF, 100 mM Na3VO4, 1 mM EDTA, 0.5% Nadeoxycholate, 0.5% NP40. For denaturing IPs, tissues were lysed in 10 mM Tris/HCl, pH7.4, 1% SDS, boiled, mixed with 9 volumes of RIPA buffer, reboiled, and centrifuged. Lysates were pre-cleared for 1 h with 30 μl protein A/G PLUS agarose beads (Santa-Cruz Biotechnology). One milliliter of cleared lysates was rotated overnight at 4°C with immunoprecipitating antibodies: 2 μg of Sox10 (rabbit, DCS Innovative Diagnostik-Systeme, cat. # SI058C01), eEF1A1 (rabbit, abcam, cat. # ab157455), EEF1A2 (rabbit, proteintech, cat. # 16091-1-AP, lot # 1370), GFP (rabbit, Abcam, cat. # ab290, lot # GR196475-1), Flag (mouse, Sigma, cat. # F1804, lot # SLBM0089V), SUMO-1 (mouse, Santa-Cruz Biotechnology, cat. # sc-5308, lot # F-1913) antibodies were used per nerve. We used rabbit anti-GFP or mouse anti-Flag antibodies as control IPs. Forty microliters of beads were added, and samples were rotated for 1 h at 4°C. Immunoprecipitates were pelleted, washed four times with RIPA buffer, eluted with 15 μl 0.1% formic acid and neutralized with 1.5 M Tris, pH 8. Six microliters of Laemmli buffer were added and samples were boiled for 10 min. Analysis was done by Western blot.

**Luciferase gene reporter assay**. Forty to fifty-percent confluent Oli-neu cells in 12-well plates were transfected either in proliferating or in differentiating medium with Fugene 6 (Promega) at a 3:1 ratio Fugene 6:DNA, according to the manufacturer's recommendations with the following modification: the DNA was incubated in Optimem for 5 min at room temperature before addition of Fugene 6. The mix Fugene 6:DNA was then incubated for 25 min at room temperature before being added to the cells. Cells were lysed 24 h after transfection in 180 μl Reporter Lysis Buffer (Promega) and assayed for luciferase activity. Twenty microliters of lysate were subjected to 2 consecutive injections of each 25 μl luciferase substrate and values were recorded after an integration time of 30 s. Efficiency of transfection was evaluated by measuring beta-galactosidase activity of a co-transfected beta-gal construct (Promega). Oli-neu cells were co-transfected with 2 μg of the luciferase construct pGL2-Mbp promoter (kind gift from Dr M. Wegner) or empty pGL2 (Promega), 2 μg of the overexpressing construct pCMV6-AC-mouse eEF1A1-turbo GFP (C-ter) or GFP backbone (Origene) or pCMV6-AC-mouse eEF1A1K273Q-turbo GFP (C-ter), and 300 ng beta-gal construct. Luciferase activity was first normalized to beta-galactosidase activity and then normalized to the empty pGL2 luciferase control (transfected with the overexpressing construct at the same ratio Fugene 6:DNA). For beta-galactosidase activity, we followed the instructions of the manufacturer (Promega). Briefly, 50 μl Assay Buffer 2x was added to 50 μl lysate, and beta-galactosidase activity was recorded after a 15-20 min incubation time at 37°C. Endogenous beta-galactosidase activity (as determined from cells only transfected with overexpressing construct at the same ratio Fugene 6:DNA) was subtracted.

**Mass spectrometry analyses**. The mass spectrometry experiment and method after IP of acetyl-lysine at 1dpl in mouse sciatic nerves treated with mocetinostat or vehicle has been previously published[41]. The data of the present study results from a different analysis of the previous experiment by Protein Match Score Summation.

For identification of eEF1A1 binding partners by mass spectrometry analysis, the collected beads were heated in SDS-PAGE loading buffer containing 1 mM DTT for 10 min at 75°C. After alkylation using 5.5 mM iodoacetamide for 10 min at room temperature the samples were centrifuged and the supernatants were loaded on 4-12% gradient gels (NuPAGE, Thermo Fisher) for protein separation. After staining, each gel lane was cut into 7 slices, the proteins were in-gel digested with trypsin (Promega) and the resulting peptide mixtures were processed on STAGE tips and analyzed by LC-MS/MS. The LC-MS measurements were performed on a QExactive Plus mass spectrometer coupled to an EasyLC 1000 nanoflow-HPLC. Peptides were separated on fused silica HPLC-column tip (I.D. 75 μm, New Objective, self-packed with ReproSil-Pur 120 C18-AQ, 1.9 μm (Dr. Maisch) to a length of 20 cm) using a gradient of A (0.1% formic acid in water) and B (0.1% formic acid in 80% acetonitrile in water). The mass spectrometer was operated in the data-dependent mode; after each MS scan (mass range m/z = 370 – 1750; resolution: 70000) a maximum of ten MS/MS scans were performed using a normalized collision energy of 25%, a target value of 1000 and a resolution of 17500. The MS raw files were analyzed using MaxQuant Software version 1.4.1.2 for peak detection, quantification, and peptide identification using a full length UniProt mouse database (April, 2016) and common contaminants such as keratins and enzymes used for in-gel digestion as reference. Carbamidomethylcysteine was set as fixed modification and protein amino-terminal acetylation, lysine acetylation, and oxidation of methionine were set as variable modifications. The MS/MS tolerance was set to 20 ppm and three missed cleavages were allowed using trypsin/P as enzyme specificity. Peptide and protein FDR based on a forwards-reverse database were set to 0.01, minimum peptide length was set to 7, and minimum

number of peptides for identification of proteins was set to one, which must be unique. The "match-between-run" option was used with a time window of 1 min.

**Reporting summary**. Further information on research design is available in the Nature Research Reporting Summary linked to this article.

## Data availability

The source data underlying Figs. 1–5, Figs. 7–9 and Supplementary Figs. 1, 4, 5, 7, 9, 11 are provided as a Source Data file. The mass spectrometry proteomics data have been deposited to the ProteomeXchange Consortium via the PRIDE [1] partner repository with the dataset identifiers PXD010552 [http://proteomecentral.proteomexchange.org/cgi/GetDataset?ID = PXD010552], PXD005383 [http://proteomecentral.proteomexchange.org/cgi/GetDataset?ID = PXD005383] and PXD017579 [http://proteomecentral.proteomexchange.org/cgi/GetDataset?ID = PXD017579]. There is no restriction on data availability. Source data are provided with this paper.

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

## Acknowledgements

*PLP-CreERT2* mice have been used in collaboration with Dr. Ueli Suter (ETH Zürich, Switzerland). We thank Dr. Jacqueline Trotter (JGU Mainz, Germany) for the Oli-neu cell line, Dr. Michael Wegner (University of Erlangen, Germany) for the Sox10 and *Mbp* promoter-luciferase constructs and for the Sox10-Nterm antibody, Sigrid Müller (Neurobiology, DBM, University Hospital Basel, Switzerland) for her technical contribution and Dr. Robin Franklin (Department of Clinical Neurosciences, MRC Cambridge Stem Cell institute, Cambridge, UK) for his advice on the analyses of remyelination in the CNS. Funding: Swiss National Science Foundation grants PP00P3_1139163, PP00P3_163759, and 31003 A _173072, grant No. P 174 from the International Foundation for Research in Paraplegia, grant from the Swiss Multiple Sclerosis Society, grant from the Olga Mayenfisch Stiftung, grant from the Forschungspool of the University of Fribourg.

## Author contributions

M.D. and C.J. conceived and designed the experiments. M.D., A.V., G.N., D.S.S., M.S., T.Z., A.H., and S.R. performed the experiments. M.D., A.V., G.N., D.S.S., M.S., J.D., M.H., and C.J. analyzed the data, D.M., T.Y., P.M., and N.S.W. contributed reagents/materials, M.D. and C.J. wrote the manuscript. All authors commented on the manuscript.

## Competing interests

Patent application # EP17174916.1 filed in Europe on 8 June 2017 and internationally on 8 June 2018 (application # PCT/EP2018/065168), applicant = University of Fribourg, Switzerland, inventor: Claire Jacob, under evaluation, title: HDAC1/2 activator for promoting and/or accelerating myelination and/or remyelination. The authors declare no other competing interest.
