## [Peer Review File · Nature Communications]

Reviewers' Comments:

Reviewer #1:

Remarks to the Author:

The paper by Duman et al. entitled "EEF1A1 Deacetylation Enables Transcriptional Activation of Remyelination" provides compelling evidence that the acetylation state of eukaryotic elongation factor 1A1 (eEF1A1) alters the subcellular localisation and abundance of Sox10, a key transcriptional regulator of both peripheral and central myelination. In Schwann cells, the authors provide biochemical and genetic evidence that acetylation of eEF1A1 promotes the nuclear translocation of eEF1A1 where it associates with and drives the nuclear export of Sox10 during Schwann cell de-differentiation. Conversely, deacetylation of eEF1A1, which they demonstrate can be mediated by HDAC2, leads to an increase in the nuclear localisation of Sox10. In a mouse model of peripheral demyelination caused by crush injury of the sciatic nerve, treatment with the HDAC2 activator, theophylline, reduced levels of acetylated eEF1A1 and increased levels of Sox10 and its transcriptional targets Krox20 and P0, resulting in increased myelin thickness. The authors also provide evidence that HDAC2 plays a similar role in regulating eEF1A1 activity in oligodendrocytes in the CNS. Pharmacological blockade or conditional deletion of HDAC1/2 in oligodendrocytes increased levels of acetylated eEF1A1 which was associated with a marked reduction in nuclear Sox10. Finally, in a mouse model of spinal cord demyelination, administration of the HDAC2 activator, theophylline, increased the levels of nuclear Sox10 and its transcriptional target, myelin basic protein, and was associated with improved remyelination efficiency.

The finding that the nuclear localisation of Sox10 is disrupted by eEF1A1 acetylation in myelinating glia is a novel and important finding which will be of broad interest in the field of myelin biology and could have therapeutic implications for the treatment of demyelinating conditions in both the PNS and CNS.

The authors have previously demonstrated that HDAC2 binds to the promoters of Sox10, Krox20 and P0 and activates their transcription (Jacob et al., Nat Neurosci, 2011). It has also been demonstrated that HDAC1 and HDAC2 promote oligodendrocyte differentiation by disrupting the interaction between beta-catenin and TCF (Ye et al., Nat Neurosci, 2009). This begs the question – to what extent does HDAC2-mediated deacetylation of eEF1A1, per se, promote the differentiation of myelinating cells, as oppose to these other known pro-myelinating effects of HDAC2. To this extent the authors have not provided a clear answer. The experiments presented in Figure 3 in which individual lysine residues within eEF1A1 are mutated to either glutamine or arginine to mimic acetylation or block acetylation respectively, provide evidence that acetylation of eEF1A1 is associated with a reduction in the levels of Sox10 in the nucleus of some cells, but the downstream consequences in terms of the effects of these manipulations of myelin gene expression is not explored. Evidence that cultured SCs or oligodendrocytes overexpressing the K273Q mutant (as compared to WT eEF1A1) exhibit reduced expression of Sox10 target genes would provide more definitive evidence that acetylation of eEF1A1 plays a key role in regulating the transcriptional program in myelinating glia.

In Figures 1 and 2, the authors examine changes in the level and subcellular localisation of acetylated eEF1A during Schwann cell de-differentiation arguing that eEF1A is deacetylated by HDAC2. However, the authors do not provide quantitative data on the total levels of eEF1A1. To argue that eEF1A acetylation, per se, is the primary mechanism that drives downstream molecular changes it would be important to also provide quantitative Western blot analysis of total eEF1A1 levels in addition to the levels of Ac-EF1A already provided, particularly in those experiments where Schwann cells are subject to genetic or pharmacological manipulations.

The authors' suggest that Tip60 is required for eEF1A1 acetylation in Schwann cells via an association with Stat3. The authors demonstrate that whilst pharmacological inhibition of Tip60 reduces acetylated eEF1A1 this does not alter the association between Stat3 and eEF1A1. What is the role of Stat3 in this complex? Is binding of Stat3 to eEF1A1 required for Tip60-mediated

acetylation of eEF1A1? The authors have demonstrated that shRNA against Stat3 reduced the association between eEF1A1 and Tip60 but have not formally shown that this reduces eEF1A1 acetylation. Experiments to clarify this point could provide greater insight into the molecular mechanisms that drive eEF1A1 acetylation.

The immunohistochemical analysis of Ac-eEF1A levels in oligodendroglia in MS lesions suggests that there are higher levels of Ac-eEF1A in OPCs within MS lesions and low levels in oligodendroglia outside the lesion. Additional evidence to support the authors' argument that HDAC2 activation could promote remyelination by reducing levels of acetylated eEF1A in MS lesions, the authors should perform additional stains for HDAC2 and Sox10. Are levels of HDAC2 and Ac-eEF1A inversely correlated in lesion-associated OPCs? Is high level Ac-eEF1A expression associated with low nuclear Sox10 (and vice versa)?

Reviewer #2:

Remarks to the Author:

In this manuscript Duman et al. investigate the mechanistic link between HDAC activity and the pro-myelination transcription factor Sox10. Performing a screen for HDAC targets they identify eEF1A1 as a protein deacetylated by HDAC1/2. They find that acetylated eEF1A1 shows an enhanced nuclear localization, where it binds to Sox10 and targets it for degradation. On the basis of a mass spec experiment and literature search, STAT3 and TIP60 are identified as a complex that promotes the acetylation of eEF1A1 (and presumably thereby degradation of Sox10). Encouragingly, in models of both PNS and CNS demyelination/remyelination, treatment of the animals with the HDAC activator theophylline enhanced repair, an effect that was lost in HDAC1/2 conditional knockouts. The group has previously published a link between HDAC activity and Sox10 levels in myelinating cells, this study provides a more substantive model of the molecular mechanisms that may link the two and provide a pathway for targeting therapeutically in demyelinating disease.

HDAC1/2 are likely to have multiple targets in myelinating cells, and the pharmacological agents used are also likely to hit multiple targets. Most of the individual experiments fail to unequivocally demonstrate that HDACs are modulating myelination through control of Sox10 levels via eEF1A1. Nevertheless, the paper is ambitious in scope and the cumulative evidence supporting their model is fairly strong. The identification of a pharmacologically targetable pathway regulating a pro-myelination transcription factor in demyelinating disease of high interest. I do have a number of concerns that reduce my enthusiasm for the paper in its current form, however:

1. The model of the paper is that acetylated eEF1A1 binds to Sox10 in the nucleus, targeting it for degradation and limiting myelination/remyelination. Although Sox10 is indisputably vital for myelination and remyelination, it is not clear to me that its levels are decreased (on a cell-by-cell level) in the context of remyelination failure or that Sox10 levels are usually rate-limiting in remyelination. Sox10 levels are already elevated post PNS crush; are the further increases seen with theophylline treatment necessarily reflective of increased HDAC activity on eEF1A1, or could they reflect changes to the density of Schwann cells in the nerve overall in the drug treated mice? Are Sox10 protein levels inversely correlated with Ac-eEF1A in the oligodendrocyte lineage cells in MS lesions?

2. Although central to the paper, the biochemical interaction between eEF1A1 and Sox10 receives little attention. Can it be linked to a specific domain in the Sox10 protein? Could targeted deletions of such a domain stabilize Sox10 in the presence of acetylated eEF1A1?

3. The evidence that STAT3 and TIP60 modulate Sox10 levels following demyelination are fairly weak. Does TIP60 inhibition or deletion lead to the predicted increase in Sox10 protein levels?

4. The acetylated eEF1A1 antibody used is raised against K41-Ac. Is it clear that acetylation at K179 and K273, which seemed to have the most effect on localization, tend to track together with acetylation at K41 and can be used interchangeably?

5. Fig. 6: Many of the myelinated axons in the remyelinating lesions appear to have myelin thickness fairly similar to the unlesioned areas, raising the concern that they are "spared" rather than remyelinated internodes. Can a g-ratio analysis be performed to confirm that the remyelinated axons display the expected increase in g-ratios?

6. Throughout much of the paper eEF1A K41-Ac is probed for without reference to total eEF1A levels. This makes it difficult to be certain whether it is acetylation or total protein levels being affected. For most of the Western blots in the paper total eEF1A would seem to be a more appropriate loading control than GAPDH.

7. MS samples: The CC1 staining appears to be so non-specific for mature oligodendrocytes as to be of limited value. I would be more positive about the MS tissue data if a quantification of Sox10 levels in oligodendrocyte lineage cells was performed in MS lesions over multiple individuals.

8. Fig 3a: It appears that levels of cytoplasmic eEF1A1 decrease rather than nuclear levels increasing with the K  Q mutations, leading to the change in ratios?

9. Theophylline has a number of targets other than HDACs, raising concerns over whether it is promoting remyelination through HDAC activation. dKO for HDAC1/2 are used to demonstrate that theophylline treatment cannot promote P0 or Sox10 levels in the absence of HDAC1/2, consistent with it acting through this pathway. Nevertheless, HDAC1/2 have been shown to have a number of targets in Schwann cells and Schwann cell development is severely disrupted in their absence. If HDACs are necessary for myelination then it is not clear that Schwann cells would be able to upregulate P0 or Sox10 in the absence of HDACs even if theophylline was not acting through them. In the absence of an obvious experimental way to address this issue, it should be acknowledged in text.

Reviewer #3:

Remarks to the Author:

Duman and colleagues report in their manuscript on a non-canonical role of the eukaryotic translation elongation factor eEF1A1 as an inhibitor of remyelination that upon Tip60-dependent acetylation gets translocated into the nucleus of Schwann cells/oligodendrocytes where it interacts with Sox10 to drag it out of the nucleus and thereby represses its transcriptional activity. The authors also provide evidence that HDACs in particular HDAC2, deacetylates eEF1A1 and thereby allows Sox10 relocation into the nucleus. They propose eEF1A1 deacetylation as a mechanism to improve remyelination in diseases such as multiple sclerosis. The presented data are both thought-provoking and innovative, of interest to others in the community and the wider field, well presented and for the most part well-supported by data. Issues to be addressed include the following:

Major:

One set of data that does not meet the overall quality standards of the manuscript is Fig. 4a,e,f. I have a tough time to see from panel a that eEF1A1 preferentially interacts with the sumoylated form of Tip60. The same quality concerns exist for panels e and f.

Additionally, the Stat3 data are very sketchy. To me various other explanations exist for the obtained results in addition to the proposed one. No data are in fact presented that directly support the existence of a ternary complex of eEF1A1, Tip60 and Stat3 as proposed in Fig. 5a. There is no evidence in the current manuscript that eEF1A1 acetylation in oligodendrocytes is also

caused by Tip60. In the absence of such evidence, generalizations should not be made or insinuated.

The other issue concerns the interpretation of the data obtained with theophylline. As the authors are probably aware, theophylline is not only a pharmacological activator of HDAC2, but also a phosphodiesterase inhibitor and an adenosine receptor antagonist among others. On a systemic level, it has anti-inflammatory and vasodilatory effects that may at least in part be mediated by activation of a very broadly expressed HDAC2. How can the authors be sure that the observed theophylline effects on PNS remyelination are primarily caused by HDAC2 activation in Schwann cells? To convince me they would need to show that all other options (other pharmacological targets of theophylline, HDAC2 in other cell types) are irrelevant, or they should replicate their results with a mechanistically different HDAC activator. If this does not exist, they should tone down their conclusions.

If I understand correctly, Fig. 7 presents data from a single multiple sclerosis patient. This is n = 1, thus not yet reproduced and clearly insufficient to draw far-reaching conclusions.

Minor:

p.3, lower half: leukodystrophies are listed among the demyelinating diseases. Should be (mostly) dysmyelinating.

Fig. 3e: The increased nuclear localization of the K to Q mutations of eEF1A1 is a bit difficult to see. A higher magnification might help.

Per journal style, a revised version should contain the uncropped Western blots and a source file for the primary data. Bar graphs should also be replaced or changed to include single data points. Statistics need to be checked.

Reviewers' comments:

Reviewer #1 (Remarks to the Author):

The paper by Duman et al. entitled “EEF1A1 Deacetylation Enables Transcriptional Activation of Remyelination” provides compelling evidence that the acetylation state of eukaryotic elongation factor 1A1 (eEF1A1) alters the subcellular localisation and abundance of Sox10, a key transcriptional regulator of both peripheral and central myelination. In Schwann cells, the authors provide biochemical and genetic evidence that acetylation of eEF1A1 promotes the nuclear translocation of eEF1A1 where it associates with and drives the nuclear export of Sox10 during Schwann cell de-differentiation. Conversely, deacetylation of eEF1A1, which they demonstrate can be mediated by HDAC2, leads to an increase in the nuclear localisation of Sox10. In a mouse model of peripheral demyelination caused by crush injury of the sciatic nerve, treatment with the HDAC2 activator, theophylline, reduced levels of acetylated eEF1A1 and increased levels of Sox10 and its transcriptional targets Krox20 and P0, resulting in increased myelin thickness. The authors also provide evidence that HDAC2 plays a similar role in regulating eEF1A1 activity in oligodendrocytes in the CNS. Pharmacological blockade or conditional deletion of HDAC1/2 in oligodendrocytes increased levels of acetylated eEF1A1 which was associated with a marked reduction in nuclear Sox10. Finally, in a mouse model of spinal cord demyelination, administration of the HDAC2 activator, theophylline, increased the levels of nuclear Sox10 and its transcriptional target, myelin basic protein, and was associated with improved remyelination efficiency.

The finding that the nuclear localisation of Sox10 is disrupted by eEF1A1 acetylation in myelinating glia is a novel and important finding which will be of broad interest in the field of myelin biology and could have therapeutic implications for the treatment of demyelinating conditions in both the PNS and CNS.

The authors have previously demonstrated that HDAC2 binds to the promoters of Sox10, Krox20 and P0 and activates their transcription (Jacob et al., Nat Neurosci, 2011). It has also been demonstrated that HDAC1 and HDAC2 promote oligodendrocyte differentiation by disrupting the interaction between beta-catenin and TCF (Ye et al., Nat Neurosci, 2009).

This begs the question – to what extent does HDAC2-mediated deacetylation of eEF1A1, per se, promote the differentiation of myelinating cells, as oppose to these other known pro-myelinating effects of HDAC2. To this extent the authors have not provided a clear answer. Answer: What we have previously shown in Jacob et al., Nat. Neurosci., 2011, is directly in line with the mechanism we are showing in the present manuscript. Indeed, we showed that HDAC2 (and also HDAC1) interacts with Sox10 to bind to the Sox10 promoter and to the regulatory regions of Sox10 target genes. HDACs cannot bind the DNA directly, they need a binding partner that is able to bind DNA, such as a transcription factor. The loci where HDAC2 is enriched on Sox10, Krox20, and P0 genes are binding sites for Sox10 on these genes. What we were missing in this previous study was the direct deacetylation target of HDAC2 that allows Sox10 to activate its target genes. In the present manuscript, we show that HDAC2 deacetylates eEF1A1 to prevent the interaction of eEF1A1 with Sox10 and Sox10 re-localization to the cytoplasmic compartment. In the previous study, we had not looked at the subcellular localization of Sox10, we just looked at total levels. To integrate the findings

of the present study to the previous ones, we have added the following part (underlined): 1/ in the Introduction “We previously showed that the two highly homologous histone deacetylases 1 and 2 (HDAC1/2) act as co-factors of Sox10³⁰⁻³³; however, in these previous studies, the deacetylation target and mechanism by which HDAC1/2 enables Sox10 activity remained elusive.”, 2/ In the Discussion “We previously showed that HDAC1/2 act as co-factors of Sox10³⁰⁻³³, but the direct deacetylation mechanism underlying this function remained unclear.”

In Jacob et al., Nat. Neurosci., 2011, we have also shown that HDAC1/2 (primarily HDAC1) prevent the activation of beta-catenin and thereby prevent the interaction of beta-catenin with TCF in early postnatal Schwann cells to maintain Schwann cell survival, a mechanism specific to this developmental stage. We also showed in this study that when the myelination process is well on going in Schwann cells, the expression levels of HDAC1 and HDAC2 decrease and the levels of active beta-catenin increase. Active beta-catenin then acts as a booster of myelination, at least in Schwann cells (also showed by other research groups. For example: Tawk et al., J Neurosci., 2011; Grigoryan et al., PNAS, 2013). In oligodendrocytes, it seems that active beta-catenin rather prevents myelination, as shown by Ye et al., Nat Neurosci., 2009, however this seems to be more complicated, as reviewed by Guo et al., Glia, 2015 and by Xie et al., Mol Neurobiol., 2014.

The mechanism of HDAC1/2 on beta-catenin is a different mechanism than the interaction of HDAC1/2 with Sox10 and deacetylation of eEF1A1 to protect Sox10 from cytoplasmic re-localization. These two different mechanisms are not mutually exclusive. In any case, the effects of HDAC1/2 on beta-catenin activation would rather inhibit the pro-myelinating effect of beta-catenin in Schwann cells, but the main effect of HDAC1/2 is to promote Schwann cell differentiation and myelination, thus the effect on Sox10 appears dominant.

The experiments presented in Figure 3 in which individual lysine residues within eEF1A1 are mutated to either glutamine or arginine to mimic acetylation or block acetylation respectively, provide evidence that acetylation of eEF1A1 is associated with a reduction in the levels of Sox10 in the nucleus of some cells, but the downstream consequences in terms of the effects of these manipulations of myelin gene expression is not explored.

Evidence that cultured SCs or oligodendrocytes overexpressing the K273Q mutant (as compared to WT eEF1A1) exhibit reduced expression of Sox10 target genes would provide more definitive evidence that acetylation of eEF1A1 plays a key role in regulating the transcriptional program in myelinating glia.

Answer: In our revised manuscript, we have addressed this point. For cultured SCs, we have quantified the percentage of Krox20-expressing cells in GFP-transfected (control), WT eEF1A1 and K273Q mutant and we found decreased % of Krox20-expressing SCs transfected with WT eEF1A1 or K273Q mutant as compared to GFP-transfected cells, and decreased % of Krox20-expressing SCs transfected with K273Q mutant as compared to WT eEF1A1-transfected SCs. We chose to quantify Krox20 expression, because *Krox20* is a Sox10 target gene that is expressed in many cultured SCs upon induction of differentiation. The results are shown in Supplementary Fig. 7.

To check the effect of wild type eEF1A1 and K273Q mutant (over)expression on oligodendrocyte differentiation, we used Oli-neu cells, which can be differentiated into mature oligodendrocytes and can be transfected, while primary oligodendrocytes are very difficult to transfect. We show that overexpression of WT eEF1A1 or expression of K273Q mutant robustly decreases the expression of Sox10 and MYRF (*Myrf* is a Sox10 target gene in oligodendrocytes) as compared to GFP-expressing cells and that K273Q mutant further

decreases Sox10 and MYRF levels as compared to WT eEF1A1-overexpressing cells, consistent with their effect in cultured SCs. The results are shown in Supplementary Fig. 11. Of note, K273Q mutant is present in both nucleus and cytoplasm of most Oli-neu cells cultured under proliferating conditions, while WT eEF1A1 is localized either exclusively in the cytoplasm or in both nucleus and cytoplasm of Oli-neu cells cultured under proliferating conditions. However, when the cells are changed to differentiation medium, K273Q mutant gets re-localized to the cytoplasm of many Oli-neu cells within a few hours (see time-course in Supplementary Fig. 11c). It is possible that Oli-neu cells find a way to get rid of K273Q mutant to maintain high Sox10 levels in the nucleus and thus to allow the cells to differentiate. In any case, even if the cells manage at some point to relocalize K273Q mutant to the cytoplasm, the levels of Sox10 and MYRF are still lower at 24h post transfection in cells transfected with K273Q mutant as compared to WT eEF1A1 or to GFP. To test whether K273Q mutant impairs the induction of differentiation in Oli-neu cells, we have carried out luciferase gene reporter assays to quantify *Mbp* promoter activity in cells either overexpressing WT eEF1A1 or expressing K273Q mutant or GFP as control. We measured luciferase activity a few minutes after the induction of differentiation. While there is a trend for decreased activity of the *Mbp* promoter in cells overexpressing WT eEF1A1, the activity of the *Mbp* promoter is very significantly decreased in cells expressing K273Q mutant as compared to GFP-transfected cells. These data are presented in Supplementary Fig. 11d.

In Figures 1 and 2, the authors examine changes in the level and subcellular localization of acetylated eEF1A during Schwann cell de-differentiation arguing that eEF1A is deacetylated by HDAC2. However, the authors do not provide quantitative data on the total levels of eEF1A1. To argue that eEF1A acetylation, per se, is the primary mechanism that drives downstream molecular changes it would be important to also provide quantitative Western blot analysis of total eEF1A1 levels in addition to the levels of Ac-EF1A already provided, particularly in those experiments where Schwann cells are subject to genetic or pharmacological manipulations.

Answer: For each IP, we had always also done the Western blot of total eEF1A1 in addition to GAPDH, except for Fig. 1c. The levels of total eEF1A1 are unchanged between the compared samples, so we can indeed argue that eEF1A1 is acetylated. In any case, we have added the Western blots for total eEF1A1 in our revised manuscript in Figs. 1b, 4h and Supplementary Fig. 1b, in addition to GAPDH. For Fig. 1c, we have quantified in Fig. 1d the levels of eEF1A1 in Control and dKO sciatic nerves at P4 to show that there is no difference in the levels of total eEF1A1 between these two groups. Thus, we can conclude that the increase in acetylated eEF1A1 levels is due to increased acetylation in the absence of HDAC1/2 and not to increased total levels of eEF1A1. However, at 3 days post lesion, there is an increase of eEF1A1 total levels, so here we have quantified the levels of Ac-eEF1A1 and have normalized to total eEF1A1 (Fig. 2b). The fold increase of Ac-eEF1A1 levels is higher than the fold increase of total eEF1A1, therefore we can conclude that besides an increase in total eEF1A1 levels, there is also a robust increase in acetylation of eEF1A1. In addition, acetylation of eEF1A1 is already increased at 1 day post lesion (Supplementary Fig. 1), while total eEF1A1 levels are not increased. Thus, acetylation of eEF1A1 occurs prior the upregulation of eEF1A1.

The authors' suggest that Tip60 is required for eEF1A1 acetylation in Schwann cells via an association with Stat3. The authors demonstrate that whilst pharmacological inhibition of

Tip60 reduces acetylated eEF1A1 this does not alter the association between Stat3 and eEF1A1. What is the role of Stat3 in this complex?

Answer: We show in Fig. 4g that knockdown of Stat3 reduces the levels of the higher molecular weight form of Tip60 that has been shown to be sumoylated Tip60 (we confirm that in our revised manuscript in Supplementary Fig. 6) and the most active form (Naidu et al., Cell Cycle, 2012; Cheng et al., Oncogene, 2008). In addition, in Fig. 4h, we show that knockdown of Stat3 impairs the interaction of Tip60 with eEF1A1. Thus, the function of Stat3 in this complex is to promote the activation of Tip60 and its binding to eEF1A1.

Is binding of Stat3 to eEF1A1 required for Tip60-mediated acetylation of eEF1A1?

The authors have demonstrated that shRNA against Stat3 reduced the association between eEF1A1 and Tip60 but have not formally shown that this reduces eEF1A1 acetylation. Experiments to clarify this point could provide greater insight into the molecular mechanisms that drive eEF1A1 acetylation.

Answer: We have added this missing part. Indeed, we show in Fig. 4i that knockdown of Stat3 impairs the acetylation of eEF1A.

The immunohistochemical analysis of Ac-eEF1A levels in oligodendroglia in MS lesions suggests that there are higher levels of Ac-eEF1A in OPCs within MS lesions and low levels in oligodendroglia outside the lesion. Additional evidence to support the authors' argument that HDAC2 activation could promote remyelination by reducing levels of acetylated eEF1A in MS lesions, the authors should perform additional stains for HDAC2 and Sox10. Are levels of HDAC2 and Ac-eEF1A inversely correlated in lesion-associated OPCs? Is high level Ac-eEF1A expression associated with low nuclear Sox10 (and vice versa)?

Answer: We have done the additional stainings of HDAC2 and Sox10 on the same MS lesions as for Ac-eEF1A. We have tried several different antibodies for HDAC2 and Sox10 and have managed to get a good staining with one HDAC2 rabbit antibody and one goat Sox10 antibody. We show that indeed the levels of HDAC2 and Sox10 are low in the lesions as compared to outside the lesions. This inversely correlates with the high levels of Ac-eEF1A in the lesions and low levels outside the lesions. These results are presented in Figure 8 and in Supplementary Figs. 14 and 15 of our revised manuscript.

Reviewer #2 (Remarks to the Author):

In this manuscript Duman et al. investigate the mechanistic link between HDAC activity and the pro-myelination transcription factor Sox10. Performing a screen for HDAC targets they identify eEF1A1 as a protein deacetylated by HDAC1/2. They find that acetylated eEF1A1 shows an enhanced nuclear localization, where it binds to Sox10 and targets it for degradation. On the basis of a mass spec experiment and literature search, STAT3 and TIP60 are identified as a complex that promotes the acetylation of eEF1A1 (and presumably thereby degradation of Sox10). Encouragingly, in models of both PNS and CNS demyelination/remyelination, treatment of the animals with the HDAC activator theophylline enhanced repair, an effect that was lost in HDAC1/2 conditional knockouts. The group has previously published a link between HDAC activity and Sox10 levels in myelinating cells, this study provides a more substantive model of the molecular mechanisms that may link the two and provide a pathway for targeting therapeutically in demyelinating disease.

HDAC1/2 are likely to have multiple targets in myelinating cells, and the pharmacological agents used are also likely to hit multiple targets.

Answer about the potential multiple targets of HDAC1/2 (similar comment as Reviewer 1):

All our studies on the functions of HDAC1/2 in Schwann cells (Jacob et al., J Neurosci., 2014; Jacob et al., Nat. Neurosci., 2011; Brügger et al., PLOS Biol., 2015; Brügger et al., Nat Comm., 2017) as well as a study from another group (Chen et al., Nat Neurosci., 2011) show that the main functions of HDAC1/2 are tightly linked to the functions of Sox10. We have also shown that HDAC1/2 (primarily HDAC1) prevent the activation of beta-catenin in early postnatal Schwann cells to maintain Schwann cell survival (Jacob et al., Nat. Neurosci., 2011), a mechanism specific to this developmental stage. We also showed in this study that when the myelination process is well on going in SCs, the expression levels of HDAC1 and HDAC2 decrease and the levels of active beta-catenin increase. Active beta-catenin seems then to act as a booster of myelination, at least in SCs (also showed by other research groups. For example: Tawk et al., J Neurosci., 2011; Grigoryan et al., PNAS, 2013). In oligodendrocytes, it seems that active beta-catenin rather prevents myelination, as shown by Ye et al., Nat Neurosci., 2009, however this is controversial and appears more complicated, as reviewed by Guo et al., Glia, 2015 and by Xie et al., Mol Neurobiol., 2014.

The mechanism of HDAC1/2 on beta-catenin is a different mechanism than the interaction of HDAC1/2 with Sox10 and deacetylation of eEF1A1 to protect Sox10 from cytoplasmic re-localization. These two different mechanisms are not mutually exclusive. In any case, the effects of HDAC1/2 on beta-catenin activation would rather inhibit the pro-myelinating effect of beta-catenin in Schwann cells, but the main effect of HDAC1/2 is to promote Schwann cell differentiation and myelination, thus the effect on Sox10 appears dominant. Two studies have reported that the transcription factor Zeb2 has major effects on Schwann cell myelination (Wu et al., Nat Neurosci., 2016; Quintes et al., Nat Neurosci., 2016), and one of them (Wu et al., Nat Neurosci., 2016) claimed that the functions of Zeb2 require the NuRD complex containing HDAC1 and HDAC2. These studies show that in the absence of Zeb2, inhibitors of differentiation such as Sox2, Id2, c-Jun, Ednrb, Hey2 are upregulated and Schwann cell proliferation is increased. In addition, rescue experiments in one of these studies on Zeb2 (Quintes et al., Nat Neurosci., 2016) show that ablation of Hey2 in the Zeb2 KO reverts the increased expression of Sox2 and Ednrb. In our study where HDAC1/2 were ablated using the Dhh-Cre mouse line (Jacob et al., Nat Neurosci., 2011) such as both studies on Zeb2, we looked at the expression of Sox2, Id2 and c-Jun and did not detect increased expression in the absence of HDAC1/2, but either no change or decreased levels. No increased level was detected for Ednrb, at least at the mRNA level. In addition, we did not detect increased proliferation in the absence of HDAC1/2 (Jacob et al., Nat Neurosci., 2011, Supplementary Figure 7). These differences show that the ablation of HDAC1/2 impairs developmental myelination through a different mechanism as for the ablation of Zeb2. Thus, even if HDAC1/2 may contribute to the functions of Zeb2, the main functions of HDAC1/2 are to drive myelination through Sox10. In addition, Sox10 re-localization to the cytoplasm, which we observe in our study in the presence of acetylated eEF1A1, is extremely likely to impair the myelination process, Sox10 being essential for myelination and remyelination. In summary, while it is possible that increasing the activity and expression of HDAC2 promotes remyelination by several mechanisms, the effect of HDAC2 in deacetylating eEF1A1 to prevent Sox10 re-localization to the cytoplasm is a major effect to promote.

Answer on the potential multiple targets of the pharmacological agents: Theophylline is a potent activator of HDAC2 at low concentrations, increasing both HDAC2 activity and expression (Barnes, Ther. Adv. Respir. Dis., 2009). At higher concentrations, Theophylline acts as an inhibitor of phosphodiesterases and as an antagonist of adenosine receptors. We have used a low concentration and dose of Theophylline in our experiments. However, to show that Theophylline increases Sox10 levels through HDAC2 and not through its other targets, we have carried out the following experiments: we have treated SCs either with 1 μ M Theophylline (low concentration) or with 1 μ M CGS 15943 (antagonist of adenosine receptors) or with 500 μ M IBMX (inhibitor of phosphodiesterases) or with 10 μ M Rolipram (inhibitor of type IV phosphodiesterases). At these concentrations, CGS 15943, IBMX and Rolipram are fully active on their targets. We show that only Theophylline treatment increases HDAC2 and Sox10 levels. CGS 15943, IBMX or Rolipram do not increase HDAC2 or Sox10 levels and some of them even decrease HDAC2 and/or Sox10 levels. These data, which are presented in Supplementary Fig. 8, thus show that Theophylline does not increase Sox10 levels through its other targets phosphodiesterases or adenosine receptors, but is very likely to induce its effect on Sox10 through its third target HDAC2. Mocetinostat is a potent inhibitor of HDAC1 (IC₅₀: 0.15 μ M) and of HDAC2 (IC₅₀: 0.29 μ M). On cells, we use it at 0.6 μ M, where it should inhibit only these two HDACs. At higher concentrations, Mocetinostat also inhibits HDAC3 (IC₅₀=1.66 μ M) and at higher concentrations it also inhibits HDAC11. Before we started to use Mocetinostat (in 2009), we have tested the effect of increasing concentrations on Sox10 and Krox20. 0.6 μ M showed to be the optimal concentration to decrease Sox10 and Krox20 expression in primary rat Schwann cells cultured under differentiation conditions, which is consistent with the phenotype of *Hdac1/2* double KO in Schwann cells (Jacob et al., Nat Neurosci., 2011). We are thus confident that Mocetinostat used at 0.6 μ M specifically targets HDAC1/2 and not other HDACs. For the in vivo experiments carried out with Mocetinostat, we have used the same dose (10 mg/kg) as in previous publications (Nural-Guvener et al., Fibrogenesis Tissue Repair, 2014) in rodents to target HDAC1 and HDAC2. There, of course, we cannot control that well the final concentration of Mocetinostat in Schwann cells in vivo, however, the other HDAC that could potentially be inhibited with this treatment is HDAC3, but HDAC3 inhibition has been shown to induce hypermyelination (He et al., Nat. Med., 2018; Rosenberg et al., Cell Rep., 2018), in contrast to HDAC1/2 inhibition and in contrast to the effect we see on Sox10 target genes with Mocetinostat treatment. Thus, we are confident that our in vivo experiments using Mocetinostat specifically target HDAC1/2.

Most of the individual experiments fail to unequivocally demonstrate that HDACs are modulating myelination through control of Sox10 levels via eEF1A1. Nevertheless, the paper is ambitious in scope and the cumulative evidence supporting their model is fairly strong. The identification of a pharmacologically targetable pathway regulating a pro-myelination transcription factor in demyelinating disease of high interest. I do have a number of concerns that reduce my enthusiasm for the paper in its current form, however:

1. The model of the paper is that acetylated eEF1A1 binds to Sox10 in the nucleus, targeting it for degradation and limiting myelination/remyelination. Although Sox10 is indisputably vital for myelination and remyelination, it is not clear to me that its levels are decreased (on a cell-by-cell level) in the context of remyelination failure or that Sox10 levels are usually rate-limiting in remyelination. Sox10 levels are already elevated post PNS crush; are the

further increases seen with theophylline treatment necessarily reflective of increased HDAC activity on eEF1A1, or could they reflect changes to the density of Schwann cells in the nerve overall in the drug treated mice?

Answer: To answer this comment, we have done several experiments and analyses.

- First, we have counted the density of Schwann cells by electron microscopy on sciatic nerves of Theophylline-treated and Vehicle-treated mice. We show in Supplementary Fig. 9a that neither the density of Schwann cells nor the density of inflammatory cells is significantly affected by Theophylline treatment.

- Second, we checked by KI67 immunofluorescence whether proliferation was affected by Theophylline treatment as compared to vehicle treatment in sciatic nerves. We show in Supplementary Fig. 9b that proliferation is not affected by Theophylline treatment.

- Third, we have counted the density of primary Schwann cells in culture treated with Theophylline or vehicle. We show in Supplementary Fig. 9c that the density of primary Schwann cells is not affected by Theophylline treatment.

Are Sox10 protein levels inversely correlated with Ac-eEF1A in the oligodendrocyte lineage cells in MS lesions?

Answer: Same question asked by Reviewer 1. We have done additional stainings of HDAC2 and Sox10 on the same MS lesions as for Ac-eEF1A. We have tried several different antibodies for HDAC2 and Sox10 and have managed to get a good staining with a HDAC2 rabbit antibody and a Sox10 goat antibody. We show that indeed the levels of HDAC2 and Sox10 are low in the lesions as compared to outside the lesions. This negatively correlates with the high levels of Ac-eEF1A in the lesions and low levels outside the lesions. These results are presented in Fig. 8 and in Supplementary Figs. 14 and 15 of our revised manuscript.

2. Although central to the paper, the biochemical interaction between eEF1A1 and Sox10 receives little attention. Can it be linked to a specific domain in the Sox10 protein? Could targeted deletions of such a domain stabilize Sox10 in the presence of acetylated eEF1A1?

Answer: To address this point, we have co-transfected HEK293 cells with WT Sox10 or Sox10-95 (2 additional AA in HMG domain) or Sox10-MIC or Sox10-HMG together with K273Q eEF1A1 or with GFP as control. To detect Sox10-MIC and Sox10-HMG, we tried 6 different antibodies and only one provided by Dr. Michael Wegner detected these two Sox10 mutants by immunofluorescence, but not by Western blot. This is why we have quantified the expression of these two mutants by immunofluorescence. The levels of WT Sox10 were quantified by Western blot and immunofluorescence and the levels of Sox10-95 were quantified by Western blot. We show that K273Q mutant leads to decreased levels of WT Sox10, Sox10-95, Sox10-MIC and Sox10-HMG, indicating that the HMG domain is sufficient for acetylated eEF1A1 to decrease Sox10 levels.

In addition to the regulation of Sox10 stability in the presence of K273Q eEF1A1 as compared to control-transfected cells, we show that Sox10 interacts with K273Q eEF1A1 also in HEK293 cells and that the additional 2 AA in the HMG domain do not impair this interaction.

Our data also suggest that the antibody provided by Dr. Wegner is efficient to immunoprecipitate Sox10-MIC and Sox10-HMG and that these two mutants also interact with K273Q eEF1A1. We have checked that three times with WT Sox10 as positive control and with different negative controls. However, since we could not verify by Western blot that Sox10-MIC and Sox10-HMG were indeed successfully immunoprecipitated, we did not

show these data in our revised manuscript. We think that the verified interaction between WT Sox10 and Sox10-95 with K273Q eEF1A1 and the decreased levels of WT Sox10 and of all mutants in HEK293 cells are enough to indicate that the HMG domain is sufficient to enable acetylated eEF1A1 to induce the decrease of Sox10 levels and to suggest that acetylated eEF1A1 interacts with Sox10 through the HMG domain. These data are presented in Supplementary Fig. 5 of our revised manuscript.

3. The evidence that STAT3 and TIP60 modulate Sox10 levels following demyelination are fairly weak. Does TIP60 inhibition or deletion lead to the predicted increase in Sox10 protein levels?

Answer: We have checked that. Indeed, TIP60 inhibition leads to increased Sox10 levels. These data are presented in Fig. 4f of our revised manuscript.

4. The acetylated eEF1A1 antibody used is raised against K41-Ac. Is it clear that acetylation at K179 and K273, which seemed to have the most effect on localization, tend to track together with acetylation at K41 and can be used interchangeably?

Answer: For this point, we treated primary Schwann cells with the HDAC1/2 inhibitor mocetinostat to increase eEF1A1 acetylation or with its vehicle for 24h, we then did an IP of eEF1A1 and analyzed by mass spectrometry the acetylated peptides of eEF1A1 (results presented in Supplementary Table 2). The results show that all peptides with acetylated K that could be detected by mass spectrometry were significantly more abundant in the mocetinostat-treated samples, indicating that several eEF1A1 K are acetylated at the same time in the absence of HDAC1/2 activity. However, mass spectrometry does not allow to detect all peptides and we could detect only 4 acetylated peptides of eEF1A1 (eEF1A1 has 20 putative acetylation sites), among which K273 was detected but K41 and K179 were not, which does not mean that these two K were not acetylated. This experiment, showing that all detected acetylated peptides are more abundant in the absence of HDAC1/2 activity suggests that acetylation of K179 and K273 could track together with acetylation of other K such as K41. The detected acetylated peptides show acetylation of K in different protein domains (K386, K219, K408 and K273, See 3D structure in Soares and Abbott, Biology Direct, 2013) and the antibody we used in Fig. 1b,c,e,f showing strongly increased levels of acetylated eEF1A in the absence of HDAC1/2 activity is directed against eEF1A acetylated at K41. These findings suggest that eEF1A1 is globally acetylated in the absence of HDAC1/2 activity and that other lysine in addition to the ones detected by mass spectrometry and by the acetylK41-eEF1A antibody that we used here are most likely simultaneously acetylated in the absence of HDAC1/2 activity.

5. Fig. 6: Many of the myelinated axons in the remyelinating lesions appear to have myelin thickness fairly similar to the unlesioned areas, raising the concern that they are “spared” rather than remyelinated internodes. Can a g-ratio analysis be performed to confirm that the remyelinated axons display the expected increase in g-ratios?

Answer: This is actually how we quantified the remyelinated axons. We measured the g-ratio of all (re)myelinated axons in the lesion site. To set the appropriate g-ratio cut off to classify axons as remyelinated or not demyelinated, we have measured the g-ratio of axons in unlesioned spinal cords in a similar area. The highest g-ratio we measured was 0.83. Thus, we decided to count axons as remyelinated when their g-ratio was equal to or above 0.835. This is indicated in the Methods section.

6. Throughout much of the paper eEF1A K41-Ac is probed for without reference to total eEF1A levels. This makes it difficult to be certain whether it is acetylation or total protein levels being affected. For most of the Western blots in the paper total eEF1A1 would seem to be a more appropriate loading control than GAPDH.

Answer: (same comment as Reviewer 1) For each IP, we had always also done the Western blot of total eEF1A1 in addition to GAPDH, except for Fig. 1c. The levels of total eEF1A1 are unchanged between the compared samples, so we can indeed argue that eEF1A1 is acetylated. In any case, we have added the Western blots for total eEF1A1 in our revised manuscript in Figs. 1d, 4h and Supplementary Fig. 1b, in addition to GAPDH. For Fig. 1c, we have quantified eEF1A1 levels in Fig. 1d in Control and dKO sciatic nerves at P4 to show that there is no difference in the levels of total eEF1A1 between these two groups. Thus, we can conclude that the increase in acetylated eEF1A1 levels is due to increased acetylation in the absence of HDAC1/2 and not to increased total levels of eEF1A1. However, at 3 days post lesion, there is an increase of eEF1A1 total levels, so here we have quantified the levels of Ac-eEF1A1 and have normalized to total eEF1A1 (Fig. 2b). The fold increase of Ac-eEF1A1 levels is higher than the fold increase of total eEF1A1, therefore we can conclude that besides an increase in total eEF1A1 levels, there is also a robust increase in acetylation of eEF1A1. In addition, acetylation of eEF1A1 is already increased at 1 day post lesion (Supplementary Fig. 1), while total eEF1A1 levels are not increased. Thus, acetylation of eEF1A1 occurs prior the upregulation of eEF1A1.

7. MS samples: The CC1 staining appears to be so non-specific for mature oligodendrocytes as to be of limited value. I would be more positive about the MS tissue data if a quantification of Sox10 levels in oligodendrocyte lineage cells was performed in MS lesions over multiple individuals.

Answer: Well, we actually found that the CC1 staining was quite good. It however also detects astrocytes in addition to mature oligodendrocytes, this is why it seems unspecific in the lesion site because of the accumulation of astrocytes in the lesions.

In any case, we have stained additional sections from the same 3 MS samples that we used to stain for Ac-eEF1A and managed to get a good staining for Sox10 and HDAC2. As already answered in above question, we found reduced levels of Sox10 and HDAC2 in the lesions as compared to outside the lesions. These results are presented in Fig. 8 and in Supplementary Figs. 14 and 15 of our revised manuscript.

8. Fig 3a: It appears that levels of cytoplasmic eEF1A1 decrease rather than nuclear levels increasing with the K  Q mutations, leading to the change in ratios?

Answer: In Fig.3a, the photos shown have been taken at the same time at the same exposure and they are photos of membranes that have been transferred at the same time in the same transfer chamber and the gels were made out of the same mix and were run at the same time in the same electrophoresis chamber. The signals detected are thus all comparable. We can see that the nuclear fraction is less loaded in the mutants as compared to WT eEF1A1, as shown by Lamin signal. Densitometry analysis shows that the nuclear fraction of the K41Q mutant is about 3 times less loaded than the nuclear fraction of WT eEF1A1, the nuclear fraction of the K179Q mutant is about 15 times less loaded than the nuclear fraction of WT eEF1A1 and the nuclear fraction of the K273Q mutant is about 5 times less loaded than the nuclear fraction of WT eEF1A1. At equal loading, the nuclear levels of the K41Q, K179Q and K273Q would thus be respectively 3x, 15x and 5x higher than

what we see on these photos. So, we can be confident that the change of ratio is at least partially due to more abundant presence of the mutants in the nuclear fraction as compared to WT eEF1A1. And if we compare the nuclear signal of GFP for the K273Q mutant to the one of WT eEF1A1, we can see that these levels are higher in the K273Q mutant as compared to WT eEF1A1, even though the loading of the nuclear fraction is 5x smaller for the mutant as compared to the WT.

In addition, the GFP signal detected by fluorescence in Fig. 3e does not indicate decreased levels.

9. Theophylline has a number of targets other than HDACs, raising concerns over whether it is promoting remyelination through HDAC activation. dKO for HDAC1/2 are used to demonstrate that theophylline treatment cannot promote P0 or Sox10 levels in the absence of HDAC1/2, consistent with it acting through this pathway. Nevertheless, HDAC1/2 have been shown to have a number of targets in Schwann cells and Schwann cell development is severely disrupted in their absence. If HDACs are necessary for myelination then it is not clear that Schwann cells would be able to upregulate P0 or Sox10 in the absence of HDACs even if theophylline was not acting through them. In the absence of an obvious experimental way to address this issue, it should be acknowledged in text.

Answer: Concerning the different targets of Theophylline, we have already discussed this point at the very beginning of Reviewer 2's comments, but here is again our answer that explains what we have done to strengthen our conclusions:

Theophylline is a potent activator of HDAC2 at low concentrations, increasing both HDAC2 activity and expression (Barnes, *Ther. Adv. Respir. Dis.*, 2009). At higher concentrations, Theophylline acts as an inhibitor of phosphodiesterases and as an antagonist of adenosine receptors. We have used a low concentration and dose of Theophylline in our experiments. However, to show that Theophylline increases Sox10 levels through HDAC2 and not through its other targets, we have carried out the following experiments: we have treated SCs either with 1 μ M Theophylline (low concentration) or with 1 μ M CGS 15943 (antagonist of adenosine receptors) or with 500 μ M IBMX (inhibitor of phosphodiesterases) or with 10 μ M Rolipram (inhibitor of type IV phosphodiesterases). At these concentrations, CGS 15943, IBMX and Rolipram are fully active on their targets. We show that only Theophylline treatment increases HDAC2 and Sox10 levels. CGS 15943, IBMX or Rolipram do not increase HDAC2 or Sox10 levels and some of them even decrease HDAC2 and/or Sox10 levels. These data, which are presented in Supplementary Fig. 8, thus show that Theophylline does not increase Sox10 levels through its other targets phosphodiesterases or adenosine receptors, but is very likely to induce its effect on Sox10 through its third target HDAC2.

In addition, we have added in the text that the absence of effect of Theophylline on Sox10 and P0 levels in the HDAC1/2 dKO suggests that Theophylline acts through HDAC1/2 to increase Sox10 and P0 levels, but could also be due to the inability of these mutants to express Sox10 and P0 with or without Theophylline.

Reviewer #3 (Remarks to the Author):

Duman and colleagues report in their manuscript on a non-canonical role of the eukaryotic translation elongation factor eEF1A1 as an inhibitor of remyelination that upon Tip60-dependent acetylation gets translocated into the nucleus of Schwann cells/oligodendrocytes

where it interacts with Sox10 to drag it out of the nucleus and thereby represses its transcriptional activity. The authors also provide evidence that HDACs in particular HDAC2, deacetylates eEF1A1 and thereby allows Sox10 relocation into the nucleus. They propose eEF1A1 deacetylation as a mechanism to improve remyelination in diseases such as multiple sclerosis. The presented data are both thought-provoking and innovative, of interest to others in the community and the wider field, well presented and for the most part well-supported by data. Issues to be addressed include the following:

Major:

One set of data that does not meet the overall quality standards of the manuscript is Fig. 4a,e,f. I have a tough time to see from panel a that eEF1A1 preferentially interacts with the sumoylated form of Tip60. The same quality concerns exist for panels e and f.

Answer: Sumoylation studies on Tip60 have been already described. Unmodified Tip60 runs at 60 kDa. Higher molecular weight forms of Tip60 from 75 to 100 kDa are sumoylated forms (Naidu et al., Cell Cycle, 2012; Cheng et al., Oncogene, 2008). To confirm that this is also the case in our study, we have added a denaturing IP of SUMO-1 and blotted for Tip60 (Supplementary Fig. 6). We show indeed three bands for Sumoylated Tip60 from 75 to 100 kDa. The two most abundant isoforms of Tip60 in our cells or the ones that are the best recognized by our Tip60 antibody are the unmodified at 60 kDa and a sumoylated form below 100 kDa. The isoform that interacts with eEF1A1 is the one that runs below 100 kDa, which is one of the sumoylated forms. Cheng et al. (2008) showed that Tip60 has increased acetyltransferase activity when sumoylated. It thus makes sense that sumoylated Tip60 interacts with eEF1A1 since it acetylates eEF1A1.

Additionally, the Stat3 data are very sketchy. To me various other explanations exist for the obtained results in addition to the proposed one. No data are in fact presented that directly support the existence of a ternary complex of eEF1A1, Tip60 and Stat3 as proposed in Fig. 5a.

Answer: In our initial submission, we showed in Fig. 4a that eEF1A1 interacts with Tip60 and with Stat3. What we were missing was that Tip60 interacts with Stat3. So, we have added these data in Fig. 4b of our revised manuscript. The reason why we had not investigated this thoroughly in our initial submission is because this has been already shown in other studies: Stat3 and Tip60 (Sapountzi et al., Int. J. Biochem. Cell Biol., 2006) are known to interact and Stat3 and eEF1A1 (Schulz et al., Biochim. Biophys. Acta, 2014) are also known to interact.

There is no evidence in the current manuscript that eEF1A1 acetylation in oligodendrocytes is also caused by Tip60. In the absence of such evidence, generalizations should not be made or insinuated.

Answer: We do not fully understand this comment. In our initial manuscript, we had nowhere stated nor suggested nor insinuated that Tip60 acetylates eEF1A1 in oligodendrocytes. We had actually not looked at this in oligodendrocytes. We only looked at the deacetylation of eEF1A1 by HDAC1/2. If Reviewer 3 thinks that some sentences need to be changed in this regard, we would be happy to do so, but we have not found anywhere anything that would suggest or insinuate that.

The reason why we have not looked at this point is actually because we think that it is not absolutely essential for our study and because it is not easy to check it. Indeed, the only possibility to detect acetylation of eEF1A1 in oligodendrocytes is to treat the cells with

mocetinostat or to genetically ablate HDAC1/2 in oligodendrocytes. Thus, to test whether Tip60 acetylates eEF1A1, we need to treat the cells with mocetinostat and the Tip60 inhibitor at the same time. In any case, we have now looked at this point and we show in differentiated primary oligodendrocytes that the Tip60 inhibitor reduces the increase of acetylated eEF1A due to mocetinostat treatment, suggesting that Tip60 also acetylates eEF1A1 in oligodendrocytes. These data are presented in Supplementary Fig. 10.

The other issue concerns the interpretation of the data obtained with theophylline. As the authors are probably aware, theophylline is not only a pharmacological activator of HDAC2, but also a phosphodiesterase inhibitor and an adenosine receptor antagonist among others. On a systemic level, it has anti-inflammatory and vasodilatory effects that may at least in part be mediated by activation of a very broadly expressed HDAC2. How can the authors be sure that the observed theophylline effects on PNS remyelination are primarily caused by HDAC2 activation in Schwann cells? To convince me they would need to show that all other options (other pharmacological targets of theophylline, HDAC2 in other cell types) are irrelevant, or they should replicate their results with a mechanistically different HDAC activator. If this does not exist, they should tone down their conclusions.

Answer: As we explained above in response to Reviewer 2's comments and in our manuscript, Theophylline is a potent activator of HDAC2 at low concentration, increasing both HDAC2 activity and expression (Barnes, Ther. Adv. Respir. Dis., 2009). At higher concentrations, Theophylline acts as an inhibitor of phosphodiesterases and as an antagonist of adenosine receptors. To show that Theophylline increases Sox10 levels through HDAC2 and not through its other targets, we have carried out the following experiments: we have treated Schwann cells either with 1 μ M Theophylline (low concentration) or with 1 μ M CGS 15943 (antagonist of adenosine receptors) or with 500 μ M IBMX (inhibitor of phosphodiesterases) or with 10 μ M Rolipram (inhibitor of type IV phosphodiesterases). At these concentrations CGS 15943, IBMX and Rolipram are fully active on their targets. We show that only Theophylline treatment increases HDAC2 and Sox10. CGS 15943, IBMX and Rolipram do not increase HDAC2 or Sox10 levels. These data show that Theophylline does not increase Sox10 levels through its other targets phosphodiesterases and adenosine receptors, but is very likely to induce its effect on Sox10 through its third target HDAC2.

HDAC2 is ubiquitously expressed, so Theophylline can increase HDAC2 levels in any cell. However only Schwann cells and oligodendrocytes are capable of expressing Sox10 and myelin proteins in the PNS and CNS. The other cells present in the sciatic nerve after crush lesion that could potentially modulate the ability of Schwann cells to remyelinate are inflammatory cells, mostly macrophages. We have counted the density of these cells in the sciatic nerve of Theophylline-treated mice as compared to vehicle-treated mice, and did not find a significant difference, indicating that Theophylline does not modulate the presence of inflammatory cells in the crushed sciatic nerve. In addition, we show that Theophylline upregulates Sox10 levels in cultures of purified primary Schwann cells where no inflammatory cell is present.

Furthermore, Theophylline does not increase HDAC2 levels in lysates of sciatic nerves of Dhh-Cre dKO P4 mice where HDAC1/2 are ablated specifically in Schwann cells (Fig. 5k), indicating that the increase of Sox10 and P0 in the sciatic nerves of P4 control mice treated with theophylline is not due to increased HDAC2 levels in other cells than Schwann cells. In addition, in our initial manuscript, we have shown that in vivo treatment of mouse pups with Theophylline increases the levels of HDAC2, Sox10 and Myelin protein zero (P0) in wild

type mice but not in Schwann cell-specific double KO HDAC1/2 (Dhh Cre;Hdac1fl/fl;Hdac2fl/fl) mice. This suggests that Theophylline needs HDAC1/2 to increase Sox10 and P0.

Theophylline (and potentially other xanthine derivatives) is the only known activator of HDAC2, there is no other one. However, we have previously shown that the overexpression of HDAC2 in primary Schwann cells leads to increased Sox10, Krox20 and P0 levels (Jacob et al., Nat. Neurosci., 2011), suggesting that increasing HDAC2 expression or activity could promote myelination.

Taken together, our data show that theophylline is extremely likely to promote the increase of Sox10 levels, myelin protein and remyelination through HDAC2. However, we have removed the statement that our data “confirm” that Theophylline induces remyelination through HDAC2.

If I understand correctly, Fig. 7 presents data from a single multiple sclerosis patient. This is n = 1, thus not yet reproduced and clearly insufficient to draw far-reaching conclusions.

Answer: We show in total the data of 3 different human MS tissue blocks (one in Fig. 7 and the two other ones in Supplementary Figs. 5-6 of our initial submission = Fig. 7-8 and Supplementary Figs. 12-15 of our revised manuscript). We initially stained 5 different ones, but only 3 showed the presence of OPCs in the lesions, so we went on with these 3. It is not very straight forward to obtain more blocks to stain. In combination with the other data presented in this manuscript, we think that the three human MS blocks we show are sufficient to at least suggest a potential beneficial effect of Theophylline treatment to promote remyelination in MS patients.

Minor:

p.3, lower half: leukodystrophies are listed among the demyelinating diseases. Should be (mostly) dysmyelinating.

Answer: We have changed this point.

Fig. 3e: The increased nuclear localization of the K to Q mutations of eEF1A1 is a bit difficult to see. A higher magnification might help.

Answer: We have done so.

Per journal style, a revised version should contain the uncropped Western blots and a source file for the primary data. Bar graphs should also be replaced or changed to include single data points. Statistics need to be checked.

Answer: We have done so in our revised manuscript.

Reviewers' Comments:

Reviewer #1:

Remarks to the Author:

The authors have provided a detailed response to the concerns raised in the initial review and have included additional data that they have incorporated into the revised version of the paper. The additional data and new experiments fully satisfy the concerns and queries that I raised previously, support the authors' hypotheses and strengthen the overall conclusions of the manuscript.

Tobias Merson

Reviewer #2:

Remarks to the Author:

The authors have generally worked to address the concerns I raised during the first round of reviews and have strengthened some key aspects of the paper. Some issues I raised were side-stepped, however:

- Are Sox10 levels really rate-limiting in remyelination (as opposed to a basal level of Sox10 being necessary but levels beyond this not making much difference)?
- dKO for HDAC1/2 are used to demonstrate that theophylline treatment cannot promote P0 or Sox10 levels in the absence of HDAC1/2, consistent with it acting through this pathway. If HDACs are necessary for myelination then it is not clear that Schwann cells would be able to upregulate P0 or Sox10 in the absence of HDACs even if theophylline was not acting through that pathway, however.

In addition, some individual data remain weak. For example:

- The mass spec ID of STAT3 as an eEF1A1 interacting protein seems to be based on an experiment with n=1 and a peptide count of 1, if I am reading the supplementary table correctly? Yes, there were prior references in the literature for interactions between STAT3 and eEF1A1, but was this the only reason to focus on STAT3 rather than Ywhaz or Capn2 or any of the other proteins identified with a low count number in the anti-eEF1A1 condition?
- The CC1 staining may be "quite good" (as stated by the authors in their rebuttal), but if it is not specific to oligodendrocytes it is not clear what its utility is in the images from MS tissue. I also concur with the other reviewer that the current qualitative analysis on a small number of blocks from one or two MS patients is insufficient to draw far-reaching conclusions from, precisely because the authors are using these data to suggest a potential beneficial effect of Theophylline treatment to promote remyelination in MS.
- I'm concerned about basing any conclusions on the Oli-neu cell line. No evidence is provided that these cells are in fact differentiating in any meaningful way (e.g., expressing robust levels of Mbp or Plp protein or assuming a differentiated morphology), and the 5-minute differentiation time-point assessed seems extremely short. Is it clear that Mbp promoter activity changes relative to its baseline in proliferative conditions in that 5 minutes? Although the authors state that primary oligodendrocytes are difficult to transfect, many groups manage to overexpress proteins in them using lentivirus or nucleofection.

Reviewer #3:

Remarks to the Author:

Dunman and colleagues have tried to answer my criticisms and did an acceptable job in improving their manuscript accordingly. Therefore I recommend publication as is.

REVIEWER COMMENTS

Reviewer #1 (Remarks to the Author):

The authors have provided a detailed response to the concerns raised in the initial review and have included additional data that they have incorporated into the revised version of the paper. The additional data and new experiments fully satisfy the concerns and queries that I raised previously, support the authors' hypotheses and strengthen the overall conclusions of the manuscript.

Tobias Merson

Answer: We are very pleased that Reviewer#1 finds that our additional data and new experiments fully satisfy his concerns and queries and support our hypotheses and strengthen our conclusions.

Reviewer #2 (Remarks to the Author):

The authors have generally worked to address the concerns I raised during the first round of reviews and have strengthened some key aspects of the paper. Some issues I raised were side-stepped, however:

- Are Sox10 levels really rate-limiting in remyelination (as opposed to a basal level of Sox10 being necessary but levels beyond this not making much difference)?

Answer: We had no intention to side-step any comment of the reviewers, we really tried to answer every comment. For this particular comment, we thought we had done the experiments requested by Reviewer#2: « Sox10 levels are already elevated post PNS crush; are the further increases seen with theophylline treatment necessarily reflective of increased HDAC activity on eEF1A1, or could they reflect changes to the density of Schwann cells in the nerve overall in the drug treated mice? » : we had answered this question by additional experiments and quantifications shown in Supplementary Figure 9 of our first revised manuscript. « Are Sox10 protein levels inversely correlated with Ac-eEF1A in the oligodendrocyte lineage cells in MS lesions? » : we had answered this question in Figure 8 and Supplementary Figures 14-15 of our first revised manuscript.

We are sorry that we did not understand that these questions asked by Reviewer#2 were not meant to answer the uncertainty as whether Sox10 levels are rate-limiting in remyelination, which was written in comment 1. together with the above questions.

To further support our claims that increased Sox10 levels as compared to basal levels can further enhance the myelination process, we refer to Figure 4, in particular Figures 4e and 4b, of Jacob et al., Nat. Neurosci., 2011, doi:10.1038/nn.2762 (see below), where we showed that in differentiated primary Schwann cells where Sox10 levels are already high and which express the major promyelinating factor *Krox20* and the myelin protein P0, the overexpression of Sox10 results in a 5-fold induction of the *Krox20* MSE, a critical enhancer for the expression of *Krox20*. In addition, overexpression of HDAC2 alone leads to a 3-fold induction of the *Krox20* MSE and co-overexpression of Sox10 and HDAC2 has a synergistic effect on the activation of the *Krox20* MSE, with a 25-fold increase. These data indicate that increased Sox10 levels in differentiated SCs can further increase the activation of its target gene *Krox20* and that this is potentiated by simultaneous increase of HDAC2 expression. In

the case of remyelination, Sox10 is upregulated at the remyelination stage to induce the remyelination program by the upregulation of Krox20 and of myelin proteins. We show here that theophylline treatment at 10dpl before remyelination starts allows to reach faster a high upregulation of Sox10, more efficient recruitment of Sox10 to its target genes *Krox20* and *P0*, faster upregulation of Krox20 and P0, faster remyelination and faster functional recovery. Upregulating Sox10 to a high level faster is very likely to induce faster remyelination and faster functional recovery. In addition, raising Sox10 to a higher expression level is also likely to help sustain the remyelination process for a longer time, as shown by the high expression of Krox20 and P0 at 30dpl and more efficient remyelination in theophylline-treated mice at 2 months post lesion. Thus, our previous work (Jacob et al., Nat. Neurosci., 2011) and current findings suggest that Sox10 levels are rate-limiting for the speed and efficiency of remyelination, and that increasing Sox10 levels and/or activity in conjunction with HDAC2 appears as a very promising strategy to accelerate and improve remyelination after lesion. In our second revised manuscript, we have added this last part of our answer in the discussion section.

Figure 4 from Jacob et al., Nat. Neurosci, 2011. (b) Luciferase fold induction of *Sox10* MCS1C, *Sox10* MCS1, *Krox20* pro, *Krox20* MSE, *P0* pro and *P0* int1 constructs by HDAC2 or HDAC1 overexpression, compared to that seen with GFP expression (set equal to 1), in differentiated RSCs. (e) Luciferase fold induction of *Sox10* MCS1C, *Sox10* MCS1C mutants $\Delta 155-585$ and $\Delta 469-470$, *Krox20* MSE and *Krox20* MSE mutant constructs by double overexpression of Sox10/GFP, Sox10/HDAC2 or Sox10/HDAC1, compared to that seen with GFP expression, in differentiated RSCs. At least three independent experiments per graph. Two-tailed Student's *t*-test, unless stated otherwise in the figure; * $P < 0.05$, ** $P < 0.01$, *** $P < 0.001$. Error bars, s.e.m.

- dKO for HDAC1/2 are used to demonstrate that theophylline treatment cannot promote P0 or Sox10 levels in the absence of HDAC1/2, consistent with it acting through this pathway. If HDACs are necessary for myelination then it is not clear that Schwann cells would be able to upregulate P0 or Sox10 in the absence of HDACs even if theophylline was not acting through that pathway, however.

Answer: For this comment, we are not absolutely sure that we understand the remaining concern of Reviewer#2. We did not at all intend to side-step this comment. At the end of this comment, Reviewer#2 wrote « In the absence of an obvious experimental way to address this issue, it should be acknowledged in text. » As requested, we had acknowledged in the text that the absence of effect of Theophylline on Sox10 and P0 levels in the HDAC1/2 dKO suggests that Theophylline acts through HDAC1/2 to increase Sox10 and P0 levels, but could also be due to the inability of these mutants to express Sox10 and P0 with or without Theophylline.

To make this point even clearer, we have now written in the manuscript :

« While theophylline had no effect on the levels of Sox10, P0 or HDAC2 in Dhh-Cre dKO mice lacking HDAC1/2 in SCs, we show that theophylline increases the levels of Sox10, P0 and HDAC2 in control littermate (Dhh-Cre negative) mice (Fig. 5k), indicating that theophylline can increase Sox10 and P0 levels also during development and suggesting that this effect may require the presence of HDAC2 and/or HDAC1, although we cannot exclude a potential inability of the Dhh-Cre dKO mutants to express Sox10 and P0 with or without theophylline or that theophylline may act through additional or other pathways to increase Sox10 and P0 in control mice. »

We hope that this extended acknowledgement as compared to our first revised version of the manuscript is now acceptable.

To strengthen this point, we had added in our first revision new experiments in Supplementary Figure 8 showing that theophylline increases the levels of HDAC2 and Sox10 and decreases the levels of Ac-eEF1A, but that the inhibitors of phosphodiesterases IBMX and Rolipram and the antagonist of adenosine receptors CGS 15943 do not. These data indicate that theophylline does not increase Sox10 levels through the inhibition of phosphodiesterases or through antagonising adenosine receptors, but is instead very likely to induce the upregulation of Sox10 through its third target HDAC2. Of note, we have previously shown that the overexpression of HDAC2 in Schwann cells leads to increased levels of Sox10, Krox20 and P0 (Fig. 3e of Jacob et al., Nat. Neurosci. 2011).

In summary, even though we agree with Reviewer#2 that the data of Figure 5k on their own are not a formal proof that theophylline induces the expression of Sox10 and P0 through HDAC2, which we have acknowledged in the text, we think that it is important to keep these data in the manuscript because it is an obvious experiment to do and we received this question several times when we presented this study during seminars, and also because it contributes to answer a question from Reviewer#3 asking whether theophylline could promote myelination through the increase of HDAC2 levels in other cells than Schwann cells present in the sciatic nerve. Since there is no HDAC2 upregulation in sciatic nerves of DhhCre-HDAC1/HDAC2 dKO mice upon theophylline treatment, we can deduce that the upregulation of HDAC2 induced by theophylline in control mouse sciatic nerves comes from Schwann cells.

In addition, some individual data remain weak. For example:

- The mass spec ID of STAT3 as an eEF1A1 interacting protein seems to be based on an experiment with n=1 and a peptide count of 1, if I am reading the supplementary table correctly? Yes, there were prior references in the literature for interactions between STAT3 and eEF1A1, but was this the only reason to focus on STAT3 rather than Ywhaz or Capn2 or any of the other proteins identified with a low count number in the anti-eEF1A1 condition?

Answer: The main reasons why we focused on Stat3 were : 1/ it had been described to interact with eEF1A1 in other cells, so the interaction with eEF1A1 was already validated in other cells, 2/ it had been described to interact with a lysine acetyltransferase (Tip60) in the cytoplasm of cells, which was fitting with a potential binding partner of eEF1A1 that could be involved in the acetylation of eEF1A1 in the cytoplasm of Schwann cells.

We started to investigate the potential involvement of Stat3 and Tip60 in eEF1A1 acetylation in January 2017. At this time, it was known that Stat3 activation was increased in Schwann cells upon sciatic nerve injury (we have added references in the main text of the manuscript).

Although we were looking for an effect of Stat3 in the cytoplasm and not as transcription factor, Stat3 activation after sciatic nerve injury indicated that it reacts to injury, which was another hint that Stat3 may be interesting for our study. In addition, Stat3 had been extensively studied in different cell types, so we were expecting to find good antibodies and reagents available for Stat3. Stat3 in combination with Tip60 was our first choice and it turned out very quickly to be very interesting for the mechanism we were focusing on. Antibodies are very expensive, so we decided to test the putative binding partners of eEF1A1 sequentially until we find interesting candidates. Among all putative binding partners of eEF1A1 detected in our mass spectrometry analysis, we had made a list of 17 candidates (which included Stat3) that are not localized in mitochondria, are not ribosomal proteins, are not secreted proteins and are not ion channels. Among these 17 candidates, only Stat3 had been shown previously to interact with eEF1A1. Cdc42 was also part of these 17 candidates that we selected and an effector of Cdc42 had been shown to interact with eEF1A1. As second candidate, we would thus have investigated Cdc42, however Cdc42 is mostly known to regulate cytoskeleton dynamics, which was not a priori fitting with the mechanism we were looking at. Among the other putative binding partners that we selected, some were not fitting with the mechanism in focus or were either very poorly described in the literature and/or not (yet) described in Schwann cells.

We focused on a low count number candidate because we were looking for a binding partner of eEF1A1 that contributes to a non-canonical function of eEF1A1. We have used mass spectrometry analyses in that case to give us some hints where to orientate our search, not to confirm interactions with binding partners. Thus, we have run 4 different IPs as n=1, but on a pool of several sciatic nerves (1 pool of 6 unlesioned sciatic nerves and 1 pool of six lesioned sciatic nerves; each pool was split into 2 equal volumes, 1 volume for IP with the GFP antibody and 1 volume for IP with the eEF1A1 antibody). We have added these details in the legend of Supplementary Table 2, where these mass spectrometry data are presented.

We think that we have now validated the interaction of Stat3 with eEF1A1 in Schwann cells and its function as a facilitator of Tip60-mediated acetylation of eEF1A1 (experiments presented in Figure 4). Therefore, we do not see the need of adding additional mass spectrometry analyses. However, our claims that Stat3 is involved in Tip60-mediated acetylation of eEF1A1 do not of course exclude that other putative binding partners of eEF1A1 detected in our mass spectrometry analysis are also part of this mechanism.

- The CC1 staining may be “quite good” (as stated by the authors in their rebuttal), but if it is not specific to oligodendrocytes it is not clear what its utility is in the images from MS tissue. I also concur with the other reviewer that the current qualitative analysis on a small number of blocks from one or two MS patients is insufficient to draw far-reaching conclusions from, precisely because the authors are using these data to suggest a potential beneficial effect of Theophylline treatment to promote remyelination in MS.

Answer: We have used the CC1 staining in combination with the Olig2 staining to differentiate outside the lesions between mature oligodendrocytes (CC1-positive/Olig2-positive) and oligodendrocyte precursor cells (CC1-negative/Olig2-positive). However, CC1 also stains for astrocytes, but the CC1 signal in astrocyte is much less bright than in mature oligodendrocytes and astrocytes are not Olig2-positive, which makes it possible to distinguish between mature oligodendrocytes and astrocytes outside the lesion. In any case, differentiating between mature oligodendrocytes and oligodendrocyte precursor cells outside the lesions was not of very high importance, so we could have removed the CC1 staining to keep only the Olig2 staining with acetylated eEF1A, if judged necessary.

Concerning the small number of samples, those are human postmortem MS samples, it is not easy to get them. To be clear, what we were showing were three different brain blocks

coming from two different MS patients, one female (two blocks in different brain regions) and one male (1 block) of different ages (58 and 75 years old) and of different MS durations (21 and 38 years). Originally, we tested 5 blocks from 3 different patients, but in two blocks of the same patient, there were no oligodendrocyte precursor cells in the lesions or too few, so we could not use them to detect the levels of acetylated eEF1A in the lesions.

Because of the small number of MS tissues and after discussion with our Editor at Nature Communications, we removed the data on human MS tissues from our manuscript.

- I'm concerned about basing any conclusions on the Oli-neu cell line. No evidence is provided that these cells are in fact differentiating in any meaningful way (e.g., expressing robust levels of Mbp or Plp protein or assuming a differentiated morphology), and the 5-minute differentiation time-point assessed seems extremely short. Is it clear that Mbp promoter activity changes relative to its baseline in proliferative conditions in that 5 minutes? Although the authors state that primary oligodendrocytes are difficult to transfect, many groups manage to overexpress proteins in them using lentivirus or nucleofection.

Answer: Before using the Oli-neu cell line for luciferase gene reporter assay, we have tested their ability to express MBP and CC1 under different culture conditions. Among 3 differentiation media that we tested, we chose the one that led to the strongest expression of MBP : we found that Oli-neu cells express high levels of MBP and CC1 already after 3 days in culture under differentiation conditions and show a complex morphology at 10 days cultured under these conditions (see below representative images of 3 independent experiments). Of note, it had been previously shown that Oli-neu cells can express myelin proteins and acquire a complex morphology in culture (Jung et al., Eur. J. Neurosci., 1995, doi : 10.1111/j.1460-9568.1995.tb01115.x).

Before deciding on the 5-min differentiation time-point for the luciferase gene reporter assay, we had tested that the activity of the *Mbp* promoter was induced in the Oli-neu cell line at the 5-min differentiation time-point as compared to proliferation - we are sorry that we forgot to make that clear in our revised manuscript - , because we expected a decrease of activity with the K273Q eEF1A1 mutant: at the 5-min differentiation time-point, the activity of the *Mbp* promoter is already increased 2 fold as compared to the basal activity in proliferation conditions. Here are below the data.

Confocal images of MBP and CC1 co-immunofluorescence and DAPI labeling in Oli-neu cells cultured in proliferation medium or for 3 days or 10 days in differentiation medium. Single optical sections are shown. Representative images of 3 independent experiments are shown.

Quantification of *Mbp* promoter activity by luciferase gene reporter assay (luciferase activity of the *Mbp* promoter normalised to the empty luciferase construct) in Oli-neu cells transfected for 24 h with the *Mbp* promoter luciferase construct and cultured in proliferation medium for 24 h (Prolif) or in proliferation medium for 23 h 55 min + 5 min in differentiation medium (5 mins diff). Unpaired two-tailed Student's *t*-tests, *p* value: ***<0.001, values=mean, error bars=s.e.m., *n*=3 independent experiments.

We have added to Supplementary Figure 11 the MBP/CC1 staining shown above in different culture conditions in Oli-neu cells, and the luciferase activity of the *Mbp* promoter in proliferation conditions and at 5 min differentiation conditions in Oli-neu cells as shown above.

We have also added a later time-point of differentiation at 3 days where we showed that Oli-neu cells express high levels of MBP and thus differentiate. These new data show that overexpression of eEF1A1 leads to reduced levels of Sox10 and MYRF, similar to what we already showed after 1 day of differentiation. In addition, we show that overexpression of eEF1A1 leads to reduced activity of the *Mbp* promoter at 3 days of differentiation.

As explained in our first revised manuscript, the K273Q mutant is localized in both nuclear and cytoplasmic compartments of virtually all Oli-neu cells cultured under proliferating conditions, but when we induce the cells to differentiate, the K273Q mutant progressively re-localizes to the cytoplasm, as shown in Supplementary Figure 11b, maybe to allow cells to express Sox10 and to induce the differentiation process. Thus, we could do the experiments on Sox10 and MYRF expression until 1 day of differentiation with the K273Q mutant but not at a later time-point where the K273Q mutant is totally relocalized to the cytoplasm. We also showed that the K273Q mutant impairs the induction of *Mbp* promoter activation at 5 min of differentiation.

For luciferase gene reporter assays, using lentiviruses is not optimal because 2 to 3 types of lentiviruses need to transduce the same cell at the same time, which is nearly impossible. We have tested that with 2 lentiviruses, one expressing DsRed and the other one expressing GFP. Over 95% of transduced cells were transduced with only one lentivirus (transduced cells expressed only DsRed or only GFP), meaning that if we use lentiviruses for luciferase gene reporter assays, most of the transduced cells with the lentivirus carrying *Mbp* promoter-luciferase will not be transduced with the lentivirus expressing eEF1A1 or K273Q mutant or the GFP control. Thus, it is highly unlikely that this method will allow to detect a difference between the effect of eEF1A1, the K273Q mutant and the GFP control on the activity of the *Mbp* promoter.

We had previously considered to do nucleofection, but a Nucleofector is extremely expensive and we could not afford it: in 2016, more than 35'000 CHF ~ 33'000 € only for the machine, then after the kits need also to be bought. In addition, when my group moved to Mainz last year, we asked a group in Mainz that had a Nucleofector and was using it for primary oligodendrocytes, however they told us that in their hands the Nucleofector is efficient to transfect siRNAs in primary oligodendrocytes, but very inefficient to transfect plasmids in primary oligodendrocytes with less than 1% transfection efficiency. In addition, we have tried different transfection reagents in primary oligodendrocytes and none in our hands resulted in more than 1% transfection efficiency. This is why we have set up the luciferase gene reporter assays with the Oli-neu cells, which transfect easily and robustly differentiate under our culture conditions. Transfection is ideal for luciferase gene reporter assays because we can very efficiently form complexes of 2-3 constructs together, which leads to almost only co-transfected cells, and very rarely to transfected cells with only one construct.

Reviewer #3 (Remarks to the Author):

Dunman and colleagues have tried to answer my criticisms and did an acceptable job in improving their manuscript accordingly. Therefore I recommend publication as is.

Answer: We are very pleased that Reviewer#3 recommends publication as is.

Reviewers' Comments:

Reviewer #2:

Remarks to the Author:

The authors have overall addressed my concerns, particularly regarding the use of the oli-neu cell line and the time-points of the assays they are performing on it.

The authors and I still have a differing perspective on whether it clear that elevation of Sox10 levels beyond a certain threshold will necessarily equate to enhanced remyelination, but this is not worth delaying publication any further over, particularly in light of the support of the other two reviewers.

REVIEWERS' COMMENTS:

Reviewer #2 (Remarks to the Author):

The authors have overall addressed my concerns, particularly regarding the use of the oli-neu cell line and the time-points of the assays they are performing on it.

The authors and I still have a differing perspective on whether it clear that elevation of Sox10 levels beyond a certain threshold will necessarily equate to enhanced remyelination, but this is not worth delaying publication any further over, particularly in light of the support of the other two reviewers.

Answer: We are very pleased that Reviewer #2 finds that we have overall addressed his/her concerns and that publication can proceed. Concerning the differing perspective of Reviewer #2 on whether it is clear that elevation of Sox10 beyond a certain threshold necessarily equates to enhanced remyelination, we have shown in our manuscript and tried to explain in our answers to the reviewers comments that the effect on remyelination is not only linked to higher levels of Sox10, but also to the timing of Sox10 upregulation and the subcellular localization of Sox10, which regulates its activity and stability. We have made this point clearer at the end of the discussion.